# Cross-linking of the endolysosomal system reveals potential flotillin structures and cargo

Jasjot Singh [1], Hadeer Elhabashy [2,3,4], Pathma Muthukottiappan [1], Markus Stepath[5,6], Martin Eisenacher [5,6], Oliver Kohlbacher [3,4,7], Volkmar Gieselmann [1] & Dominic Winter [1] ✉

Lysosomes are well-established as the main cellular organelles for the degradation of macromolecules and emerging as regulatory centers of metabolism. They are of crucial importance for cellular homeostasis, which is exemplified by a plethora of disorders related to alterations in lysosomal function. In this context, protein complexes play a decisive role, regulating not only metabolic lysosomal processes but also lysosome biogenesis, transport, and interaction with other organelles. Using cross-linking mass spectrometry, we analyze lysosomes and early endosomes. Based on the identification of 5376 cross-links, we investigate protein-protein interactions and structures of lysosome- and endosome-related proteins. In particular, we present evidence for a tetrameric assembly of the lysosomal hydrolase PPT1 and a heterodimeric structure of FLOT1/FLOT2 at lysosomes and early endosomes. For FLOT1-/FLOT2-positive early endosomes, we identify >300 putative cargo proteins and confirm eleven substrates for flotillin-dependent endocytosis, including the latrophilin family of adhesion G protein-coupled receptors.

Lysosomes, the central lytic organelles of mammalian cells, are of crucial importance for cellular homeostasis. This is underscored by the detrimental consequences resulting from impairment of lysosomal function: mutations in genes encoding lysosomal proteins are causative for a group of ~70 rare and frequently devastating diseases, so-called lysosomal storage disorders (LSDs). Moreover, lysosomal dysfunction has been demonstrated in a number of more common conditions, including neurodegenerative diseases and cancer[1,2].

In addition to the long-known role of lysosomes in the degradation of intra- and extracellular substrates, more recent findings place them at the center of metabolic signaling. The major player in this context is the mammalian target of rapamycin complex 1 (mTORC1), whose activity is regulated at the lysosomal surface. This regulation is mediated by several protein complexes located in/at the lysosomal membrane, which integrate the activity of major signaling pathways, as well as the concentration of various metabolites[3]. Furthermore, protein complexes were shown to play a role in other lysosome-related processes, such as their transport, direct interaction with other cellular compartments, gene regulation, immunity, cell adhesion/migration, and plasma membrane repair[4].

For most of these functions, protein–protein interactions (PPIs) at the lysosomal membrane play a decisive role. Nutrient sensing and activation of mTORC1 are regulated by the interaction of at least 30 individual proteins[5], and lysosomal motility is controlled by the

[1]Institute for Biochemistry and Molecular Biology, Medical Faculty, University of Bonn, 53115 Bonn, Germany. [2]Department of Protein Evolution, Max-Planck-Institute for Developmental Biology, 72076 Tübingen, Germany. [3]Institute for Bioinformatics and Medical Informatics, University of Tübingen, 72076 Tübingen, Germany. [4]Department of Computer Science, University of Tübingen, 72076 Tübingen, Germany. [5]Medical Proteome-Center, Medical Faculty, Ruhr-University Bochum, 48801 Bochum, Germany. [6]Medical Proteome Analysis, Center for Protein Diagnostics, Ruhr-University Bochum, 48801 Bochum, Germany. [7]Institute for Translational Bioinformatics, University Hospital Tübingen, 72076 Tübingen, Germany. ✉e-mail: dominic.winter@uni-bonn.de

reversible association to microtubules through dynein and kinesin by several adaptor/scaffold complexes, such as BLOC1-related complex (BORC)[6]. The core feature of lysosomes, their acidic pH, is maintained by the 1.25 MDa vacuolar-type ATPase (V-ATPase) complex, which consists of 35 subunits (17 unique proteins), and catalyzes the transport of protons across the lysosomal membrane[7]. Delivery of certain lysosomal proteins is achieved by members of the homotypic fusion and protein sorting (HOPS)[8], the class C core vacuole/endosome tethering (CORVET)[9], as well as adaptor protein (AP) complexes, and the endosomal sorting complex required for transport (ESCRT) mediates repair of lysosomal membranes[10].

Also, for interactions of lysosomes with other organelles, protein complexes play an essential role. This includes fusion events with cargo delivery vesicles such as endosomes, phagosomes, and autophagosomes[4], exocytosis at the plasma membrane[11], or direct interactions with the endoplasmic reticulum[12], the Golgi apparatus[13] peroxisomes[14], RNA granules[15], and mitochondria[16]. The latter facilitates, for example, the exchange of small molecules and was shown to regulate events such as mitochondrial fusion and fission[4].

The majority of protein complexes that facilitate these processes are poorly characterized, and novel members/interactors are continuously being identified. Given the central role of lysosomes in metabolic regulation, and the high number of cellular structures they interact with, it is highly likely that several functionally important interactors of lysosomal proteins are still unknown.

Although structural data are available for a number of lysosomal luminal proteins and complexes in/at its membrane, three-dimensional information is still lacking for a significant fraction of the lysosomal proteome. The majority of existing structural data originates from crystallography experiments, heavily relying on affinity-purified proteins, or fragments thereof, from pro- or eukaryotic overexpression systems and their crystallization in vitro. The applicability of these structures to the in vivo situation remains therefore, in some instances, questionable[17].

A promising avenue to identify interactions of lysosomal proteins, and to obtain insights into their structure under physiological conditions, is chemical cross-linking in combination with mass spectrometry-based proteomics (XL-LC-MS/MS)[18]. In cross-linking experiments, a chemical linker forms covalent bonds between certain amino acids such as lysine. In subsequent MS analyses, these bonds are identified, providing direct proof for the interaction of proteins within a certain distance constraint, defined by the type of cross-linker[19]. This allows for the identification of PPIs, and hence localization, with high confidence. Compared to other commonly used approaches, such as immunoprecipitation (IP), proximity labeling[20], or lysosome enrichment[21], such data provide superior spatial evidence. Furthermore, the distance constraints of the cross-linker can serve as a basis for the molecular modeling of proteins and their complexes. This allows supplementing well-established techniques such as nuclear magnetic resonance (NMR), X-ray crystallography, or cryo-electron microscopy (cryo-EM), compensating for missing/incomplete data, and validating predicted protein structures[22].

So far, XL-LC-MS/MS experiments have been performed for samples of varying complexity, ranging from individual proteins[23] and multi-subunit complexes[24,25] to whole organelles[26,27] and cell/tissue lysates[28,29]. A dedicated analysis of lysosomal proteins by cross-linking has not been performed to date, which is certainly related to the fact that lysosomal proteins are of low abundance (estimated at 0.2% of cellular protein mass)[30].

In this work, we present a XL-LC-MS/MS dataset of the endolysosomal compartment of HEK293 cells, applying the MS-cleavable cross-linker disuccinimidyl sulfoxide (DSSO) to lysosome-/early endosome-enriched fractions. We present an interaction map of lysosomal proteins, of which we verify selected PPIs by co-IP and validate/extend existing protein structures. Based on the cross-linking data and integrative modeling, we further propose a tetrameric assembly of PPT1 and a heterodimeric structure for FLOT1/FLOT2. Finally, by affinity purification and MS analysis of FLOT1/FLOT2-positive early endosomes, we investigate the putative cargo of these vesicles and confirm selected candidates by co-IP and immunostaining.

## Results

### Cross-linking mass spectrometry analysis of lysosome-enriched fractions

In mammalian cells, the majority of lysosomal proteins are of relatively low abundance, and whole-cell XL-LC-MS/MS studies typically cover only a fraction of the lysosomal proteome (Supplementary Fig. 1a). A way to overcome this limitation is lysosome enrichment, which was shown by us to increase signal intensities for certain lysosomal proteins up to 100-fold relative to whole-cell lysates[31]. Accordingly, we enriched lysosomes by superparamagnetic iron oxide nanoparticles (SPIONs, Fig. 1a),[32,33] and established cross-linking conditions for lysosome-enriched fractions utilizing the MS-cleavable cross-linker DSSO[34]. Due to the limited membrane permeability of DSSO[35], we cross-linked lysosomes both in an intact (IT) and disrupted (DR) state, and determined optimal reaction conditions by silver staining and western blotting (Supplementary Fig. 1b, c). Subsequently, we enriched lysosomes from 384 plates of HEK293 cells across six biological replicates, and assessed lysosomal intactness, recovery, and enrichment (Fig. 1b, c). Using a non-cross-linked fraction of each sample, we acquired an LC-MS/MS reference dataset. In total, we identified 4181 proteins, of which 474 were assigned the term "lysosome" based on GO terms and UniProt classifiers in >3 runs, indicating an excellent performance of lysosome enrichment[36] (Supplementary Data 1 and 2). To assess the quantitative distribution of lysosomal proteins in our samples, we utilized these data to estimate absolute protein abundances by intensity-based absolute quantification (iBAQ)[37]. This revealed a threefold overrepresentation of lysosomal protein abundance relative to the whole dataset (Fig. 1d and Supplementary Data 2).

We cross-linked lysosome-enriched fractions in both the IT and DR state (three replicates each), followed by their proteolytic digestion, strong cation-exchange (SCX) peptide fractionation, and analysis by LC-MS/MS (Fig. 1a). Analysis of the XL-LC-MS/MS dataset with XlinkX[38] resulted in the assignment of 6580 cross-link spectral matches, originating from 4294 cross-linked peptides at a false discovery rate (FDR) of 5% (Supplementary Fig. 1d–f and Supplementary Data 3). Out of the 2467 unique residue-to-residue cross-links, 524 identifications (270 intralinks between different residues of the same protein and 254 inter-links between two different proteins) originated from 111 proteins assigned to the lysosomal compartment, while the remaining cross-links were identified for proteins which are currently not connected to lysosomes, presenting potentially novel interaction partners (Fig. 1e, Supplementary Fig. 1g, h and Supplementary Data 3). Interestingly, only 25% of cross-links were found both in the IT and the DR state, with the latter contributing a larger fraction to the dataset, further demonstrating the limited membrane permeability of DSSO (Fig. 1f). A similar distribution was observed for the cross-links identified for non-lysosomal proteins contained in the dataset (Supplementary Fig. 1d), and we observed a better reproducibility in cross-link identification for the DR state (Supplementary Fig. 2f). Strikingly, while cross-link spectral matches (CSMs) of cytosolic proteins were almost equally distributed between both conditions, 91% of the dataset's CSMs assigned to lysosomal luminal proteins were identified in samples cross-linked in the DR state (Fig. 1g).

As lysosomal proteins are expressed at a dynamic abundance range encompassing three orders of magnitude[36], we further correlated CSMs, peptide spectral matches (PSMs), and iBAQ values. Even though higher abundant proteins tended to yield more CSMs, we did

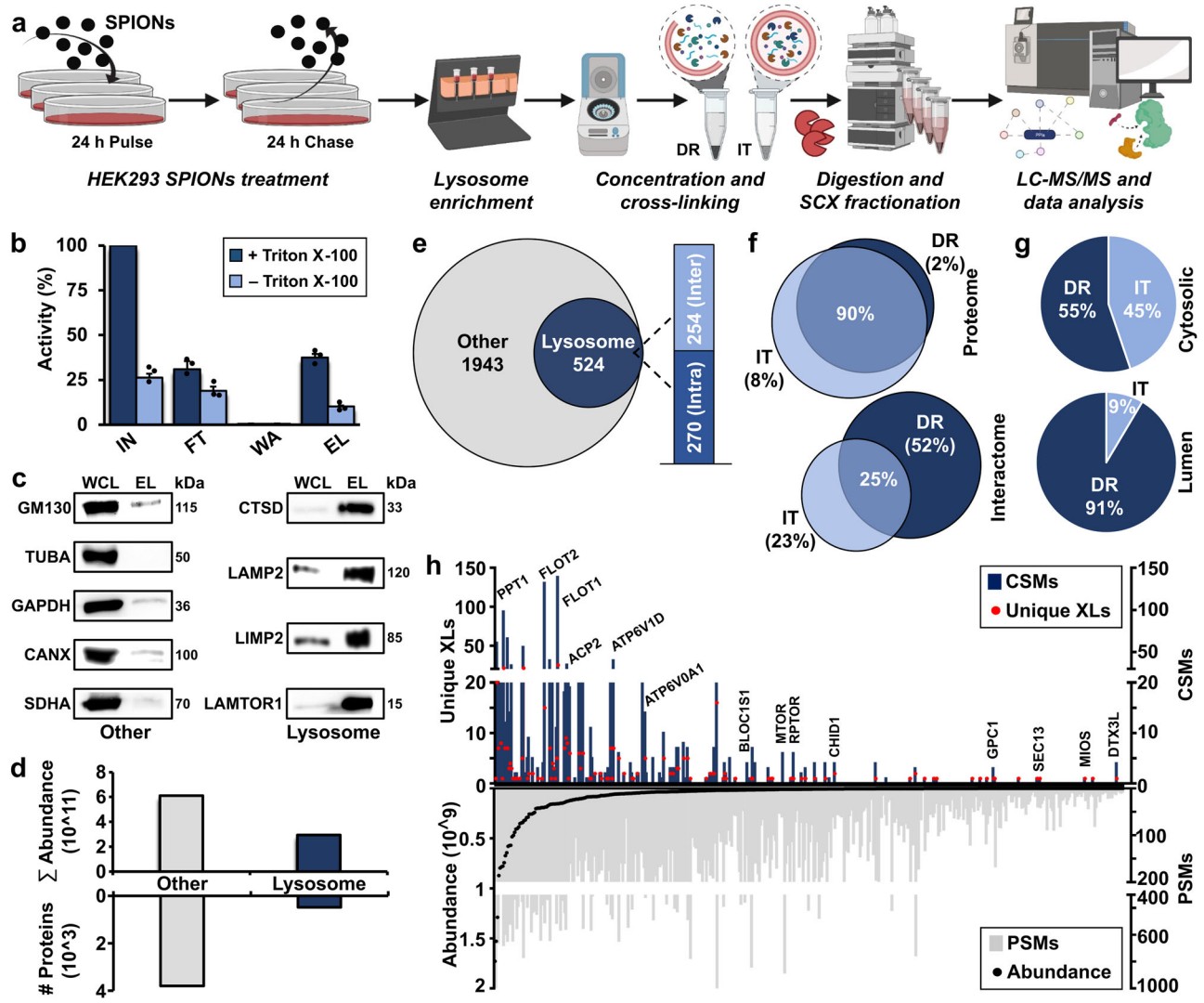

**Fig. 1 | Cross-linking mass spectrometry analysis of lysosome-enriched fractions. a** Experimental workflow for the XL-LC-MS/MS analysis of lysosome-enriched fractions. Created with BioRender.com. **b** Normalized β-hexosaminidase activities for individual fractions from lysosome enrichment by SPIONs. Data are presented as mean values + SD (*n* = 3, biologically independent samples over three independent experiments). **c** Western blot analysis of lysosome-enriched fractions for contamination by other organelles (*n* = 2). Lysosome: lysosomal proteins (CTSD, LAMP2, LIMP2, and LAMTOR1). Other: Golgi apparatus (GM130), cytoskeleton (TUBA), cytosol (GAPDH), endoplasmic reticulum (CANX), and mitochondria (SDHA). **d** Summed iBAQ abundances for proteins identified in lysosome-enriched fractions in ≥3 replicates. **e** Classification of unique cross-linked residue pairs. **f** Proteins detected in non-cross-linked lysosome-enriched fractions (proteome),

and unique lysosomal cross-linked residue pairs (interactome) for DR and IT samples. **g** Localization of CSMs for 68 lysosomal proteins cross-linked in the DR and IT state. Cytosolic: proteins located at the cytosolic face of the lysosomal membrane; Lumen: lysosomal luminal proteins. **h** Correlation of cross-link identification and protein abundance for lysosomal proteins. CSMs and PSMs represent summed values of the analysis (*n* = 6, biologically independent samples over six independent experiments (3× DR and 3× IT)). SPIONs superparamagnetic iron oxide nanoparticles, DR disrupted, IT intact, SCX strong cation-exchange, IN input, FT flow through, W wash, EL eluate, WCL whole-cell lysate, iBAQ intensity-based absolute quantification, XLs cross-links, CSMs cross-link spectral matches, PSMs peptide spectral matches. Source data are provided as a Source Data file.

not identify a strong correlation between cross-link identification and protein abundance, showing that our dataset also covered proteins of low expression levels (Fig. 1h, Supplementary Fig. 1i and Supplementary Data 2 and 3). When we compared the average iBAQ abundance for proteins involved in intra- and inter-links, we observed a tendency towards identifying more intralinks in higher abundant proteins, which was less pronounced for lysosomal proteins (Supplementary Fig. 1j, k).

Finally, we investigated the distribution of CSMs across a shortlist of lysosomal proteins/complexes[21]. Most categories showed an equal distribution between DR and IT samples, with the exception of proteins involved in lysosomal substrate degradation (93%), heat-shock proteins (70%), and annexins (75%), for which more cross-links were annotated in the DR sample (Supplementary Fig. 1l).

## Investigation of lysosomal protein–protein interactions

Based on all inter-links in the dataset, we constructed a network of 1008 proteins engaged in 1023 interactions, of which 254 involved lysosomal proteins (Fig. 2a, Supplementary Fig. 2a–c and Supplementary Data 3). Comparison to known interactions revealed an overlap of ~30%, confirming the validity of our dataset. While 34% of interactions of non-lysosomal proteins were included in STRING, only 26% of potential lysosomal PPIs have been reported previously (Fig. 2b). We classified lysosomal PPIs based on the interacting subcellular compartment, revealing an overrepresentation of nuclear and cytoplasmic/cytoskeletal proteins (Fig. 2c). With respect to lysosomal and lysosome-associated proteins, we identified the highest numbers of PPIs for the V-ATPase, the flotillins, mTORC1, and the syntaxins (Fig. 2d, e and Supplementary Fig. 2d, e).

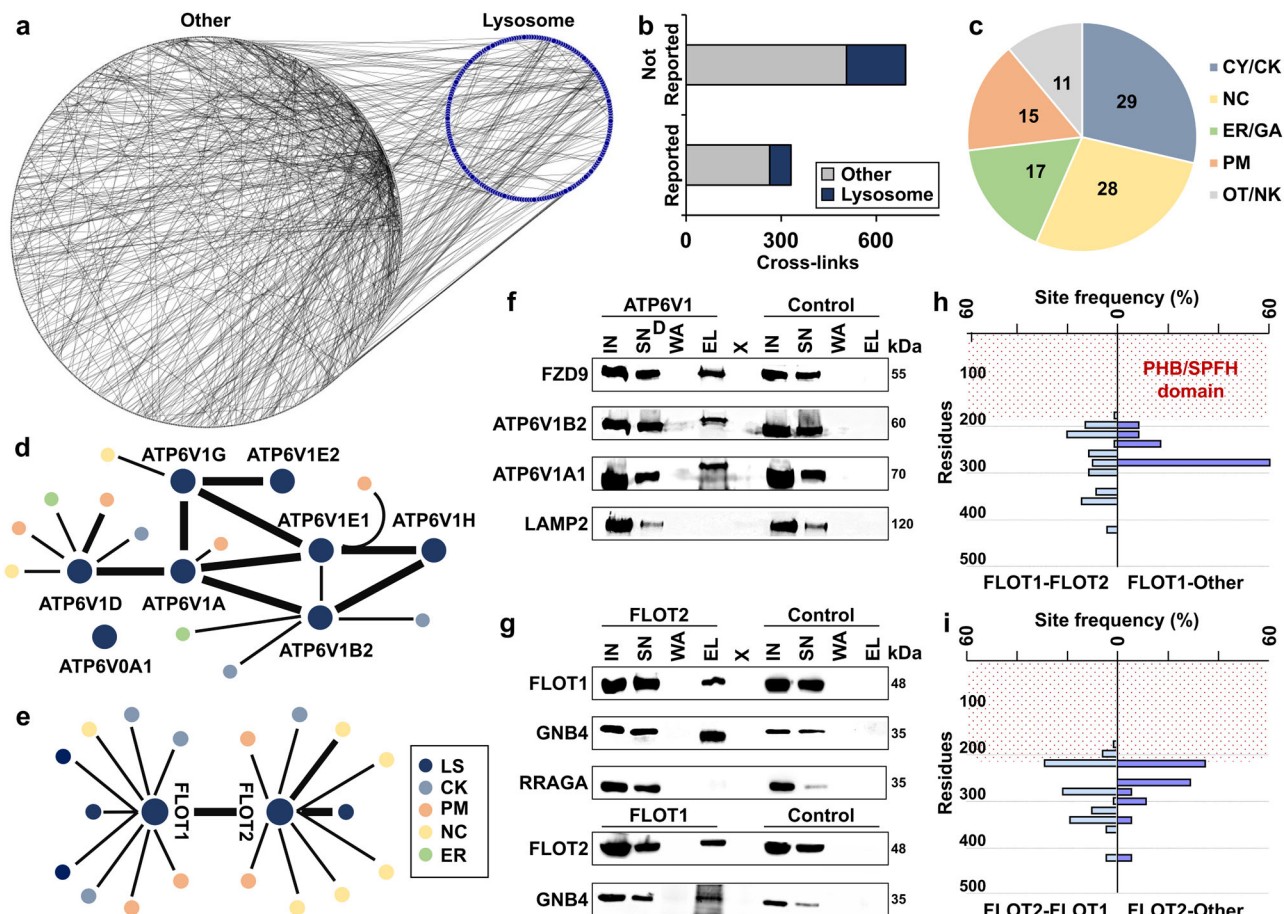

Fig. 2 | **The human lysosomal interactome. a** PPIs identified in the XL-LC-MS/MS dataset. Lysosomal proteins (blue dots), non-lysosomal proteins (gray dots), and PPIs (gray lines) are indicated. **b** Matching of PPIs to the STRING database. **c** Numbers of proteins from distinct subcellular localizations interacting with lysosomal proteins. **d, e** Interaction networks of the V-ATPase (**d**) and the flotillin (**e**) complex. **f** Co-IP of ATP6V1D and FZD9 (*n* = 2). ATP6V1B2 and ATP6V1A1 are known members of the V-ATPase complex; LAMP2 is a lysosomal membrane protein. Control: empty beads. **g** Co-IPs of FLOT1, FLOT2, and GNB4 (*n* = 2). RRAGA is a lysosomal membrane-associated protein. Control: empty beads. **h, i** Site frequency distribution for identified FLOT1 (**h**) and FLOT2 (**i**) cross-links. Site frequency represents the percentage of cross-links detected in bins of 20 residues each. The region indicated by red dots represents the PHB domain. CY cytoplasm, CK cytoskeleton, NC nucleus, ER endoplasmic reticulum, GA Golgi apparatus, PM plasma membrane, LS lysosome, OT others, NK not known, IN input, SN super-natant, W wash, EL eluate, X empty lane. Source data are provided as a Source Data file.

For the V-ATPase complex, we detected most PPIs for the D subunit of its soluble V1 part (ATP6V1D). This may be related to the capability of V1 to dissociate from the lysosomal membrane-embedded V0 part, which was shown to be involved in the regulation of V-ATPase activity[39], exposing the D subunit to interactions. The cross-link of ATP6V1D with frizzled 9 (FZD9), a member of the WNT signaling pathway, sparked our interest. FZD9 was shown to be sorted to late endosomes/lysosomes after its internalization by endocytosis[40], and the ATP6AP2 subunit of the V-ATPase was reported to interact with FZD8[41]. We, therefore, investigated the validity of this PPI by co-IP, confirming both the observed interaction of ATP6V1D with FZD9, as well as interaction with other subunits of the V-ATPase complex (Fig. 2f).

We further investigated the interaction of FLOT1, FLOT2, and GNB4, for which we identified an inter-link with FLOT2. We were able to co-IP FLOT1 and FLOT2, which are known to form heterooligomers[42], as well as GNB4 with its direct interactor FLOT2 as well as with FLOT1, indicating binding of GNB4 to FLOT1/FLOT2 heteromeric assemblies (Fig. 2g).

As we observed 18 different PPIs for FLOT1 and FLOT2 in total, we further investigated their distribution across both proteins. While FLOT1/FLOT2 inter-links were detected across most of the regions predicted to form a helical structure (amino acids 193–365

and 213–362 for FLOT1 and FLOT2, respectively)[43], the interaction with other proteins occurred almost exclusively in confined sections of <100 amino acids (Fig. 2h, i). The fact that we found FLOT1/FLOT2 inter-links across the whole region of the proteins shows that no sequence-dependent bias towards the cross-link reaction or detectability exists. Therefore, the presence of the majority of PPIs with proteins other than FLOT1/FLOT2 at a distinct section of the proteins suggests the presence of FLOT1/FLOT2 interaction hotspots.

## Structural integration of cross-linker distance constraints suggests a tetrameric assembly of PPT1 in vivo

Cross-links between different amino acids provide distance information (the length of DSSO links is ~35 Å) that is helpful to validate (or infer) protein structures[44]. Initially, we used TopoLink[45] to match 161 unique cross-links to the resolved structures of 34 lysosomal and lysosome-associated proteins, confirming the validity of our dataset. The remaining 64 cross-links assigned to lysosomal proteins could not be integrated, as the respective regions have not been resolved yet. We matched these cross-links either to homology models based on available PDB structures from other organisms using SWISS-MODEL[46], or to the AlphaFold Protein Structure Database[47] (Fig. 3a, Supplementary Fig. 3 and Supplementary Data 4).

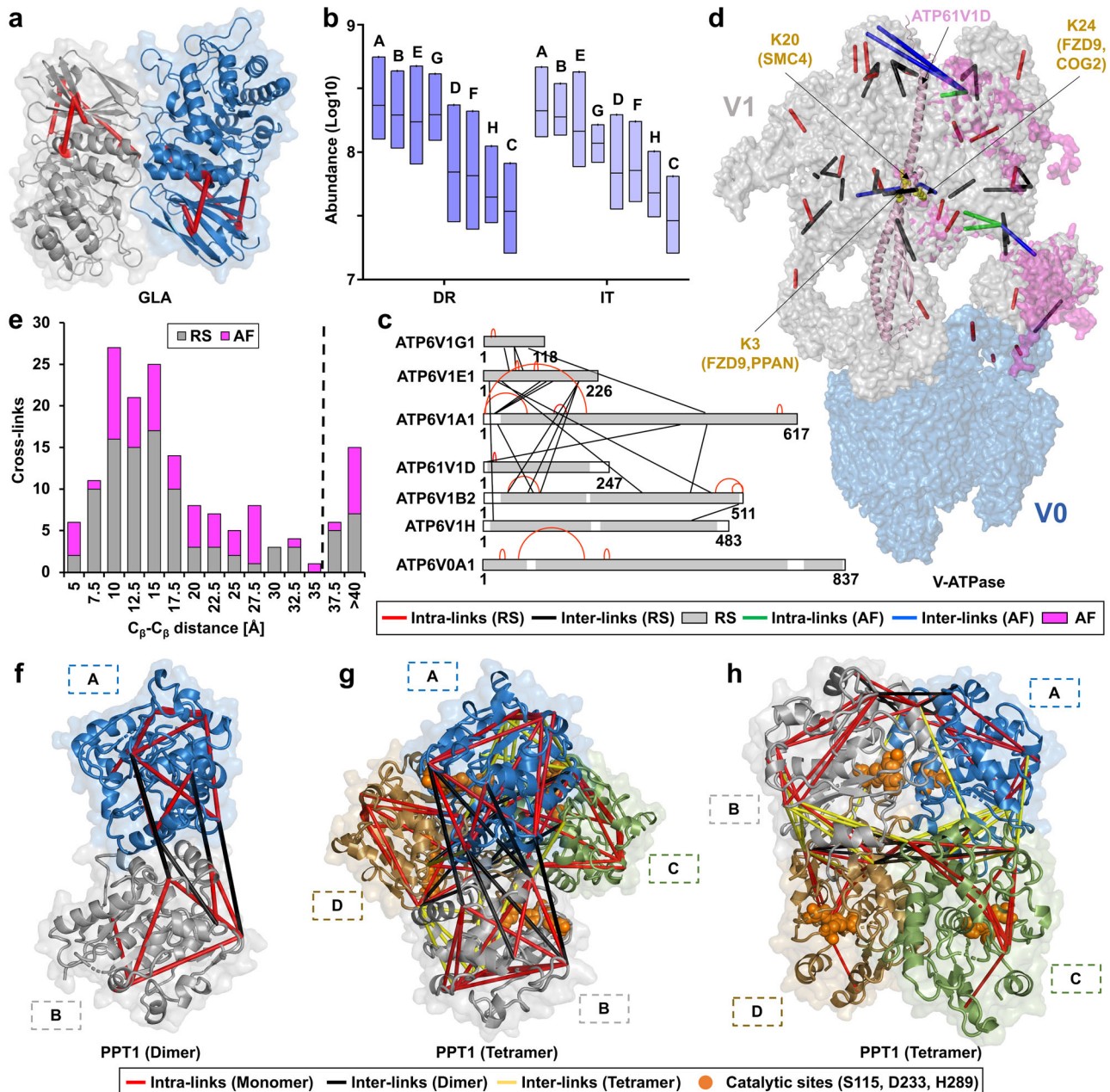

**Fig. 3 | Cross-link-based structural refinement of the V-ATPase and PPT1 complex. a** Matching of cross-links to the crystal structure of lysosomal alpha-galactosidase A (GLA). **b** Average iBAQ abundances for individual V-ATPase subunits based on the analysis of lysosome-enriched fractions in a non-cross-linked state. Each box depicts the interquartile range (IQR, the range between the 25th and 75th percentile, median = line). **c** Cross-links identified for individual V-ATPase subunits. Structurally resolved regions are colored gray, and unresolved regions are white. **d** Refined structure of the V-ATPase complex. Identified cross-links were integrated into the V-ATPase cryo-EM structure[7]. Missing regions were supplemented with predicted structures from AlphaFold2 based on the identified cross-links. **e** Distances for mapped cross-links to crystal structures and AlphaFold2 models for lysosomal proteins, bin size: 2.5 Å. **f** Reported homodimeric human PPT1 structure (PDB identifier 3GRO). **g, h** Tetrameric PPT1 model representing the most favorable energetic state and fulfilling DSSO's distance constraints for all 18 cross-links. A, B, C, and D indicate individual subunits. DR disrupted, IT intact, RS resolved structure, AF AlphaFold2. Source data are provided as a Source Data file.

The complex for which we identified the highest number of cross-links was the V-ATPase, a 35-mer assembly of 17 unique proteins[7], of which 13 were covered in our dataset. While the soluble V1 part (eight different proteins) yielded 26 cross-links, only three were identified for the membrane-embedded V0 section (seven different proteins). After confirmation of the V1 part's correct stoichiometry through the individual subunit's iBAQ values in the non-cross-linked dataset ($A_3B_3E_3G_3D_1F_1C_1H_1$) (Fig. 3b and Supplementary Data 2), we mapped the identified cross-links to a recently published structure determined by cryo-EM[7]. Out of the 29 unique cross-links identified, 21 could

readily be integrated into the published structure, while 8 originated from regions that were so far structurally not resolved (Fig. 3c). For these sections, we integrated the predicted full-length protein models by AlphaFold2, aligning them with the published V-ATPase subunit structures in the complex, based on the identified cross-links (Fig. 3d). In this combined model, we cover ~95% of the V-ATPase sequence, and 90% of cross-links fulfill DSSO's distance constraints. The remaining three cross-links were inter-links of the A and E subunit of the V1 part, which are known to undergo conformational changes during ATP hydrolysis[7].

Regarding mTORC1, we identified most cross-links for proteins related to Ragulator, a lysosomal membrane-associated complex that is crucial for mTORC1 activity[3]. For the N-terminal region of LAMTOR1, we identified three cross-links of the same lysine residue with different amino acids of LAMTOR3. Two of them violated DSSO's distance constraints relative to the crystal structure obtained in the presence of the Ragulator-associated RAG GTPases[48], indicating cross-linking of an alternative state, possibly representing Ragulator in the absence of RAG GTPases (Supplementary Fig. 3j). We further identified three intralinks from RRAGA that originated from the same lysine residue at its C-terminus, cross-linked to three different amino acids (Supplementary Fig. 3k). This could be related to high flexibility of this region, which is in accordance with the fact that it could not be covered in a previous crystallization study[49].

Of the 161 lysosomal cross-links identified, 21 exceeded DSSO's distance constraint (Fig. 3e and Supplementary Data 4). Surprisingly, nine of them originated from PPT1, a member of the palmitoyl protein thioesterase family. The current PPT1 crystal structure (PDB identifier 3GRO) resembles a homodimeric assembly of two identical subunits (Fig. 3f). Strikingly, while all eleven PPT1 intralinks fulfilled the distance constraints of this structure, the inter-links exceeded them, indicating the possibility of an alternative oligomerization state. This possibility is in line with a previous study detecting the maximal enzymatic activity of PPT1 at a complex size of >100 kDa[50]. We, therefore, used HAD-DOCK2.2 to perform restraint-based docking for the monomeric subunits of PPT1, extracted from 3GRO. This resulted in the prediction of a tetrameric PPT1 model, which fulfills the distance constraints for all 18 PPT1 cross-links (Fig. 3g, h).

## Proposal of a heterodimeric FLOT1/FLOT2 model featuring extended alpha-helical domains

We identified the highest number of cross-links for the two members of the flotillin family, FLOT1 and FLOT2, which were also over-represented in the proteomic dataset of the lysosome-enriched fraction (Fig. 1h and Supplementary Data 2). FLOT1 and FLOT2 are lipid raft-associated proteins which are present in nearly every type of vertebrate cell, and are highly conserved among organisms[43]. We confirmed their co-enrichment with lysosomes by western blotting (Fig. 4a) and colocalization by immunostaining (Fig. 5a and Supplementary Fig. 5a), which is in agreement with previous EM studies detecting FLOT1 at the lysosomal surface[51,52]. It is well-established, that FLOT1 and FLOT2 form heterodimers with a 1:1 stoichiometry[53], but only partial structural information is available from NMR analyses of the N-terminal region of mouse Flot2 (PDB identifier 1WIN). Other experimental data on FLOT structures are not available, as purification of the full-length proteins is problematic[54].

Our dataset contains 29 unique cross-links for FLOT1 and FLOT2, including 11 intra- and 22 inter-links (Supplementary Data 3). In the first step, we predicted individual secondary structures for both FLOT1 and FLOT2 using PSIPRED (Fig. 4b)[55]. For the protein N-termini, this resulted in a cluster of beta-sheets, which is in accordance with their sequence homology to the stomatin/PHB/flotillin/HflK/C (SPFH) domains and the Flot2 NMR structure. The middle section features an extended α-helical region that is interrupted once in the case of FLOT2, while the C-terminus forms one beta-sheet and several short helices for both proteins. We further calculated coiled-coil probabilities for the helical regions using PCOILS[56], with window sizes of 14, 21, and 28 amino acids (Fig. 4b). Dependent on the region of both proteins, windows of 14 and 28 amino acids delivered the best results, with slightly different patterns for FLOT1 and FLOT2. Matching our data to full-length structural models from the AlphaFold Protein Structure Database showed excellent agreement, and all identified cross-links confirmed the predicted structures.

Subsequently, we built heterodimeric models of FLOT1, FLOT2, and their interaction using ColabFold[57], a variant of AlphaFold2. For

additional post-modeling validation with experimental data, we mapped the cross-linking restraints on the five resulting models. All generated models satisfied all experimentally identified restraints based on the cross-links, increasing our confidence in these models. The resulting models feature closely aligned highly similar structures for both flotillins. In particular, they feature a globular N-terminal region consisting of SPFH domains (residues 1–162 of FLOT1 and FLOT2), a central linear α-helical region (residues 163–341 of FLOT1 and 163–350 of FLOT2), and a C-terminal α-helical coiled-coil structure (residues 342–427 of FLOT1 and 351–426 of FLOT2). The SPFH domains of both FLOT1 and FLOT2 present with antiparallel β-sheets, with six repeats each, and four partially exposed α-helices, forming an ellipsoidal-like globular domain (Fig. 4c and Supplementary Fig. 4). Based on the fact that the major helices were not interrupted, and that the C-terminus was in its most compact state, we selected model four, which was also supported by all FLOT1/FLOT2 cross-links detected (Fig. 4c and Supplementary Fig. 4c).

We highlighted several structural features in our model which are known for both flotillins (Fig. 4c). The S-palmitoylation and N-myristoylation sites, which are crucial for the membrane-association of both flotillins[58], are located at the SPFH domain's membrane interfaces. The tyrosine phosphorylation sites, which were shown to be crucial for flotillin-mediated endocytosis and FLOT1/FLOT2 interaction[59,60], are surface-exposed and in proximity to a basic motif (HQR) on the respective other flotillin. Moreover, the PDZ3 domains of both flotillins strongly co-localize, forming a combined feature, and the EA-rich motifs, which were predicted to mediate flotillin oligomerization[43,61], are distributed along the length of the central α-helical region[62]. Interestingly, the putative interaction hotspots (Fig. 2h, i) locate around the major bend observed in this structure, containing one tyrosine residue (Y238/Y241) in its center. These residues are located in highly conserved sequence motifs (A-X-A-X-L-A-pY-X-L-Q with X: [D/Q or E/Q]), possibly presenting a regulatory switch for FLOT1/FLOT2 PPIs by tyrosine phosphorylation. To further confirm the validity of our model and the presence of the predicted interaction hotspot, we generated deletion mutants lacking this entire region (amino acid residues 200–300) for both FLOT1 and FLOT2. We co-transfected both constructs and immunoprecipitated FLOT1/FLOT2 followed by the detection of GNB4 (Fig. 2g). Deletion of the interaction hotspot prevented binding of GNB4, indicating that this region is crucial for its interaction with FLOT1/FLOT2 (Fig. 4d).

## Flotillins assemble in similar higher-order structures at lysosomes and endosomes

While FLOT1 was detected previously at the cytosolic face of lysosomes by EM[51,52], assemblies of both FLOT1 and FLOT2 were only shown for the plasma membrane or early endosomes, where they play a role in clathrin-independent endocytosis[63,64]. In line with these findings, we detected FLOT1 and FLOT2 to localize at the plasma membrane and to partially co-localize with lysosomes of HeLa and HEK293 cells (Fig. 5a and Supplementary Fig. 5a). In these analyses, we also observed numerous FLOT1/FLOT2 puncta, which did not co-localize with lysosomes, presenting putative FLOT1/FLOT2-positive endosomes.

In order to elucidate if the structural assembly of FLOT1 and FLOT2 differs between early endosomes and lysosomes, we performed cross-linking of early endosome-enriched fractions. To increase the number of FLOT1-/FLOT2-positive early endosomes, we co-transfected cells with FLAG-tagged FLOT1 and FLOT2, as their overexpression was shown to increase the number of FLOT-positive endosomes[42,53]. After confirmation of correct FLOT distribution utilizing FLAG-, GFP-, and mCherry-tagged versions of FLOT1 and FLOT2 (Supplementary Figs. 5b and 6a), we established SPIONs pulse-chase conditions for the enrichment of early endosomes (Fig. 5b). Analysis of the early endosome-enriched fraction by western blot and LC-MS/MS verified

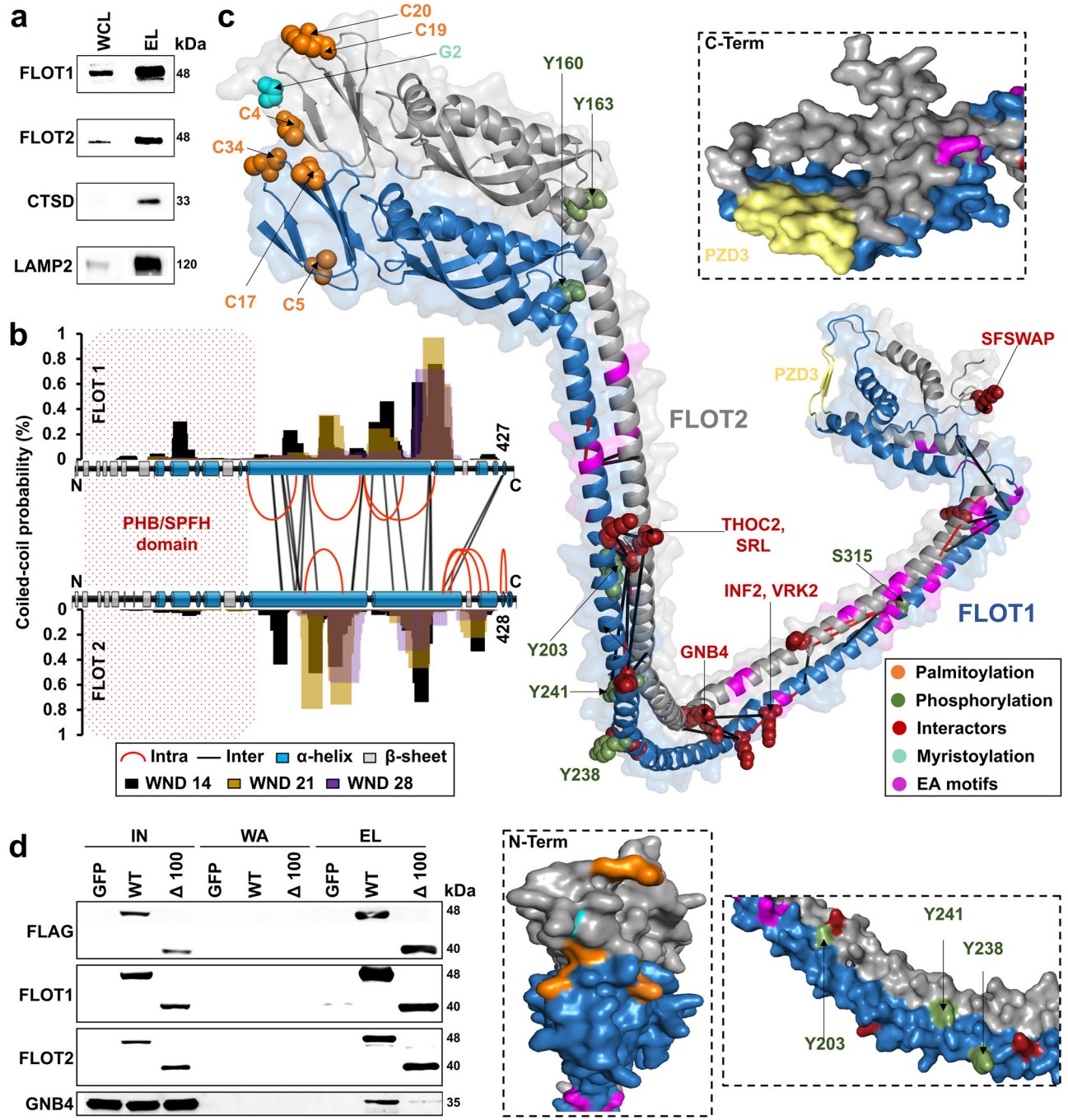

**Fig. 4 | Proposed model for a heterodimeric FLOT1-FLOT2 complex. a** Western blot analysis of lysosome-enriched fractions ($n = 3$). CTSD is a lysosomal luminal and LAMP2 a lysosomal transmembrane protein. **b** Identified cross-links, predicted secondary structures (PSIPRED), and coiled-coil probabilities (PCOILS) for FLOT1 and FLOT2. **c** Heterodimeric model of FLOT1-FLOT2 interaction (ColabFold). The model satisfies the distance constraints of all cross-links. **d** FLAG-IP from HEK293 cells transfected with either full-length FLAG-FLOT1 and FLAG-FLOT2 or versions lacking amino acid residues 200–300 (Δ100) ($n = 2$). WCL whole-cell lysate, EL eluate, PHB prohibitin homology domain, SPFH stomatin/PHB/flotillin/HflK/C domain, WND window, E glutamic acid, A alanine, PZD3 postsynaptic density protein-95/discs large/zonula occludens-1, IN input, WA wash, WT wild-type, Δ100 mutant with a deletion of 100 amino acids (residues 200–300 of FLOT1/FLOT2). Source data are provided as a Source Data file.

the presence of marker proteins such as EEA1, the clathrin chains CLTA, CLTB, and CLTC, as well as the RAB-GTPases RAB5, RAB11, and RAB14. Furthermore, we detected FLOT1 and FLOT2, while markers for other organelles were depleted (Fig. 5c and Supplementary Data 5). We then performed enrichment of early endosomes from FLOT1/-FLOT2-FLAG overexpressing HEK293 cells, established their cross-linking, performed MS sample preparation followed by SCX fractionation, and investigated them by LC-MS/MS (Fig. 5d and Supplementary Fig. 6b–f).

Importantly, western blot analysis of FLOT1/FLOT2 aggregation in response to different amounts of DSSO showed that the concentration we used for cross-linking of lysosome-enriched fractions does not result in flotillin over-cross-linking (Supplementary Fig. 5c, d), further supporting the validity of these data. In total, we identified 1081 cross-links from 414 unique proteins (Fig. 5e, Supplementary Fig. 6f and Supplementary Data 6). This early endosome dataset contains 15 unique cross-links for FLOT1 and FLOT2, which all matched to our

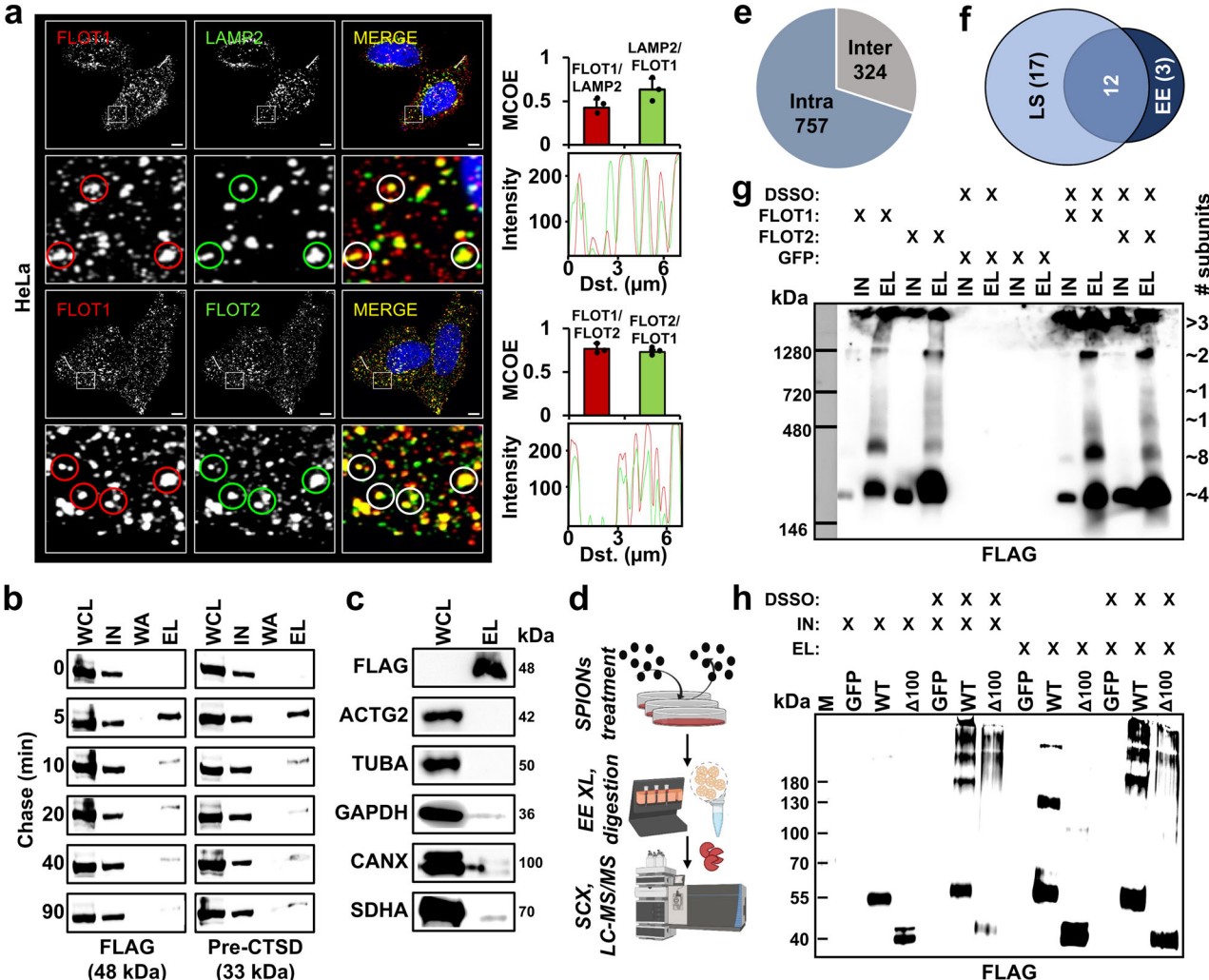

**Fig. 5 | Investigation of higher-order flotillin assemblies in lysosome- and early endosome-enriched fractions. a** HeLa cells were stained with antibodies for FLOT1, FLOT2, and LAMP2 followed by microscopy imaging ($n = 3$). Lower panels display a zoom-in of the regions indicated in the full picture. Mander's coefficients show the average degree of overlap of one protein population relative to another. Data are presented as mean values + SD ($n = 3$, cells examined over one experiment). Profile plots indicate the degree of colocalization for individual vesicles. **b** Analysis of FLOT1/FLOT2 in early endosomes ($n = 1$). Following pulsed SPIONs treatment, endosomes were enriched at the indicated chase times. Pre-CTSD serves as a marker for early endosomes. **c** Western blot analysis of early endosome-enriched fractions with marker proteins for different subcellular compartments ($n = 1$): Cytosol (ACTG2 and GAPDH), cytoskeleton (TUBA), endoplasmic reticulum (CANX), and mitochondria (SDHA). **d** Experimental workflow for early endosome

enrichment and XL-LC-MS/MS analysis. Created with BioRender.com. **e** Overview of the cross-linking dataset obtained from early endosome-enriched fractions. **f** Overlap of unique FLOT1/FLOT2 cross-links for XL-LC-MS/MS analyses of early endosome- and lysosome-enriched fractions. **g** Western blot analysis of BN-PAGE-separated FLAG-IP eluates with/without cross-linking by DSSO ($n = 5$). IN/EL refers to individual FLAG-IP fractions. **h** Western blot analysis of SDS-PAGE separated FLAG-IP ELs with/without DSSO cross-linking for WT and FLOT1/FLOT2 residue 200–300 deletion mutants (Δ100) ($n = 3$). Scale bar = 5 μM; WCL whole cell lysate, IN input, WA wash, EL eluate, MCOE Mander's coefficient, Dst. distance, LS lysosomes, EE early endosomes, XL cross-linking, SPIONs superparamagnetic iron oxide nanoparticles, M marker, WT wild type. Source data are provided as a Source Data file.

predicted FLOT1-FLOT2 heterodimer within DSSO's distance constraints, and from which 80% overlapped with the cross-linking dataset from lysosome-enriched fractions (Fig. 5f). These data indicate that early endosome- and lysosome-localized flotillins assemble in a similar way.

It has been shown previously, that flotillins form higher-order assemblies[61,63]. We, therefore, performed blue native (BN)-PAGE experiments to investigate the size distribution of FLOT1/FLOT2 structures in a native and a cross-linked state (Fig. 5g). These analyses revealed that a significant amount of both FLOT1 and FLOT2 migrates at a range corresponding to a tetrameric assembly, while smaller fractions migrated at sizes consistent with higher-order structures exceeding 1 MDa. The cross-linking samples, which were generated with the same reaction conditions as the early

endosome and lysosome experiments, presented with the same complex sizes, indicating that the cross-link data represent the native state.

Based on the combined 32 unique cross-links from the lysosome- and early endosome cross-linking datasets, we further attempted to model the tetrameric structure using either restraint-based docking or AlphaFold-Multimer. This did, however, not result in a convincing structural model. We further investigated in how far the deletion of the putative interaction hotspot influences the assembly of higher-order structures by co-transfection of both FLOT1/FLOT2 deletion constructs followed by IP, cross-linking and SDS-PAGE. Western blot analyses of these samples indicate that the absence of the interaction hotspot also influences the formation of higher-order assemblies migrating at sizes >180 kDa (Fig. 5h).

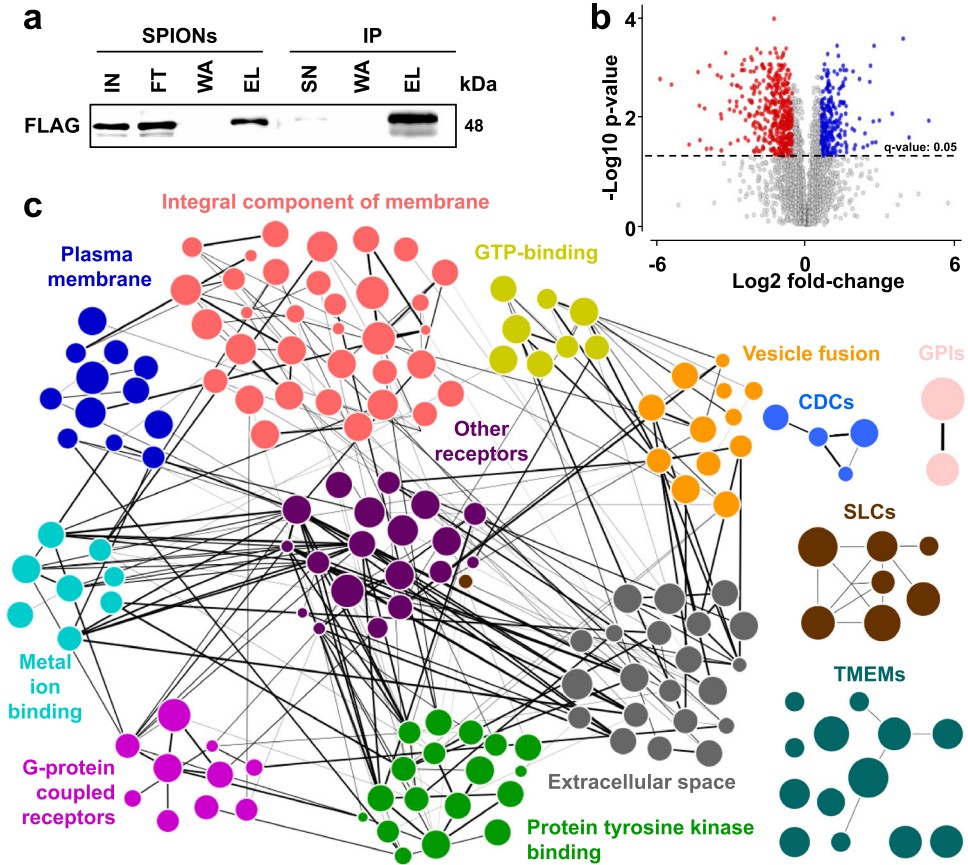

**Fig. 6 | Identification of potential FLOT1-/FLOT2-positive early endosome cargo. a** Western blot analysis of FLAG-FLOT1/FLOT2 in early endosome-enriched fractions (SPIONs) and intact endosome-IP fractions (*n* = 1). **b** Data-independent acquisition (DIA)-based protein abundance fold-change ratios of SPIONs/SPIONs +IP fractions (*n* = 3, biologically independent samples over three independent experiments). Significant differentially regulated proteins are indicated (cut-offs: *q*- value: <0.05, fold change: >1.5). **c** STRING-based PPI analysis of selected protein categories overrepresented in the SPIONs+IP fraction. Node size corresponds to the protein abundance and line thickness to the PPI confidence score. SPIONs superparamagnetic iron oxide nanoparticles, IP immunoprecipitation, IN input, SN supernatant, FT flow through, WA wash, EL eluate, PPI protein–protein interaction. Source data are provided as a Source Data file.

## Analysis of flotillin-endosome cargo reveals an over-representation of membrane proteins and receptors

Flotillins have been proposed as defining structural components of an endocytic pathway independent of clathrin and caveolin[53,65], and were shown to co-localize with early endosomes[64]. In agreement with these findings, we detected their co-enrichment with endosomes (Fig. 5b, c). To investigate the putative cargo of FLOT1-/FLOT2-positive early endosomes, we combined SPIONs enrichment of early endosomes from HEK293 cells overexpressing FLAG-tagged FLOT1/FLOT2 and immunoprecipitation of FLAG-positive intact vesicles from these samples (Fig. 6a). Subsequently, we analyzed the resulting fractions by data-independent acquisition (DIA) LC-MS/MS and performed label-free quantification (Supplementary Fig. 7a–d). Based on 5089 proteins quantified across all conditions, we were able to define protein populations that were enriched/depleted in FLOT1-/FLOT2-positive endosomes relative to the total cellular pool of early endosomes (Fig. 6b and Supplementary Data 7). Importantly, the FLOT1-/FLOT2-depleted population of early endosomes contained the clathrin chains CLTA and CLTC, EEA1, and the endosome-related GTPases RAB5 and RAB11, confirming a separation of clathrin- and flotillin-containing early endosomes. We confirmed these findings by co-immunostaining of EEA1 with endogenous FLOT1/FLOT2 in HeLa and HEK293 cells as well as in cells and transfected with FLOT1-GFP, showing that FLOT1-/ FLOT2-positive vesicles present a distinct population from EEA1-positive endosomes (Supplementary Figs. 7e, f and 8).

The enriched population, which presents potential cargo of FLOT1-/FLOT2-positive early endosomes, consists of 328 proteins (Supplementary Data 7). Of these, 189 were also identified in the lysosome-enriched fractions (Supplementary Data 2), possibly indicating their lysosomal destination. GO-enrichment analysis revealed a significant overrepresentation of several categories related to organelle and membrane proteins (Supplementary Fig. 7g), which is in agreement with a potential role of flotillins in the endocytosis and vesicular transport of plasma membrane proteins, demonstrated previously e.g., for NPC1L1[66,67]. We subsequently performed STRING analyses to sub-classify potential cargo proteins (Fig. 6c). This further revealed enrichment of different receptors types and members of the transmembrane protein (TMEM) family and the solute carrier (SLC) family of transporters (Supplementary Data 7).

For a subset of putative FLOT-positive endosome cargo proteins, we performed FLOT1 co-IPs to investigate a potential interaction. In total, we were able to IP eleven proteins with FLOT1 (Fig. 7a, c), encompassing eight receptors, two membrane proteins (PLSCR1 and VANGL1), and one protein interacting with a receptor (SMAD3). Among these, one group of receptors, the latrophilins, sparked our interest, as we detected all three members of this G protein-coupled receptor subfamily (LPHN1, LPHN2, and LPHN3) to be significantly enriched in FLOT1-/FLOT2-positive endosomes (Fig. 7b). We were able to co-IP all latrophilins with both FLOT1 and FLOT2, implicating their interaction

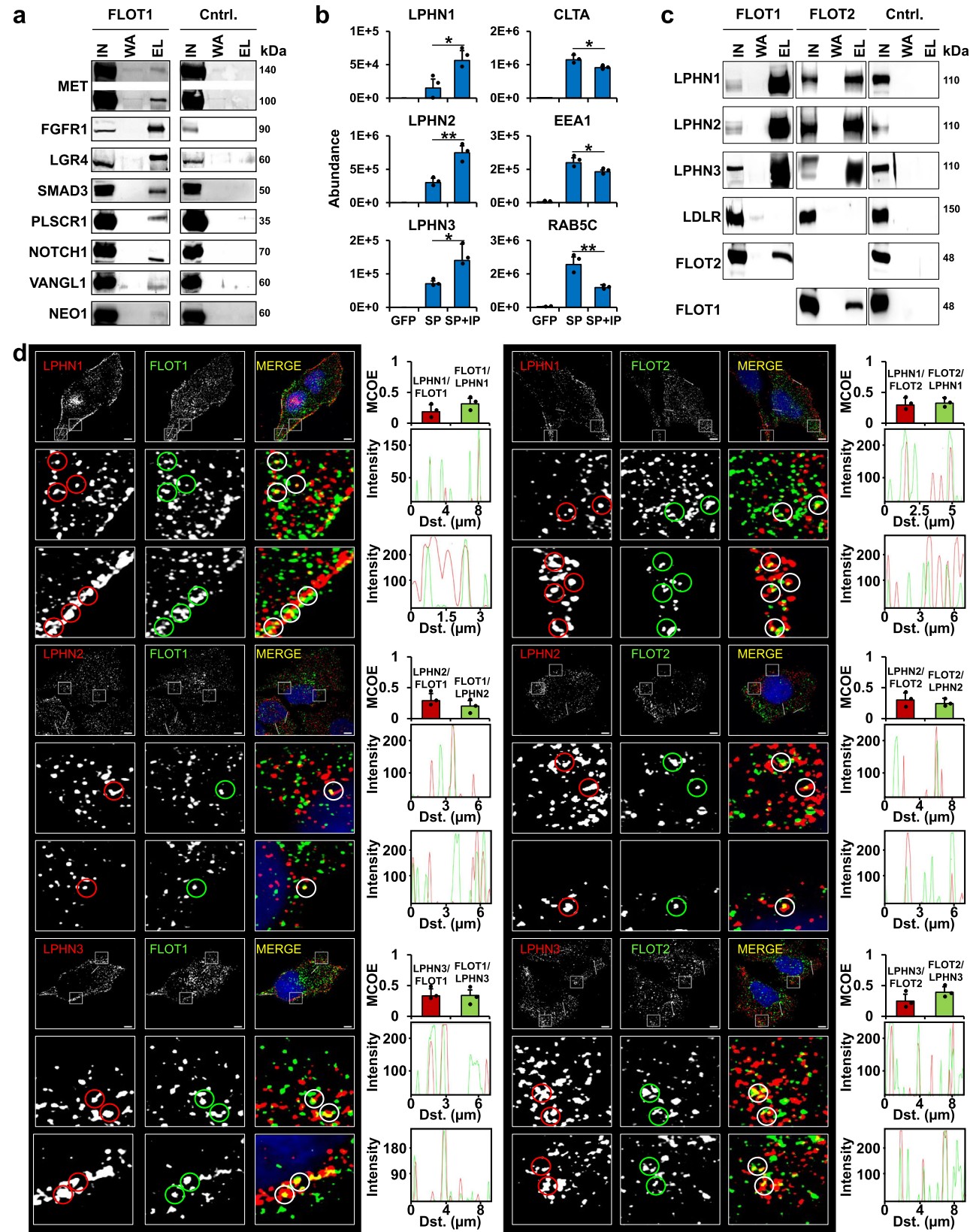

with heterooligomeric FLOT structures or both monomers (Fig. 7c). We furthermore confirmed their colocalization to FLOT1-/FLOT2-positive endosomes by immunofluorescence in HeLa and HEK293 cells (Fig. 7d and Supplementary Fig. 9), providing additional evidence that the latrophilins present a potential cargo for flotillin-mediated endocytosis.

## Discussion

In this study, we present a XL-LC-MS/MS analysis of lysosome-enriched fractions. In comparison to whole-cell analyses, we detected significantly higher numbers of cross-links for proteins reported previously to be related to the lysosome. This was especially the case for bona fide lysosomal proteins, which are localized in/at the organelle[68].

**Fig. 7 | Confirmation of possible FLOT1-/FLOT2-positive early endosome cargo proteins. a** Co-IP of FLOT1 with possible cargo proteins detected by DIA-LC-MS/MS analysis of FLOT1-/FLOT2-positive early endosomes ($n = 1$). **b** Average DIA protein abundance for LPHN1, LPHN2, LPHN3, CLTA, EEA1, and RAB5C. Data are presented as mean values + SD ($n = 3$, biologically independent samples over three independent experiments). Significance based on Student's unpaired two-sided $t$ test; range: *$P < 0.05$, **$P < 0.01$; exact $P$ values: LPHN1 (0.022), LPHN2 (0.004), LPHN3 (0.044), CLTA (0.047), EEA1 (0.035), RAB5C (0.008). **c** Co-IP of FLOT1 and FLOT2 with LPHN1, LPHN2, and LPHN3. LDLR serves as marker for plasma membrane and clathrin-mediated endocytosis. **d** Immunostaining of HeLa cells for FLOT1 and FLOT2 in combination with LPHN1, LPHN2, and LPHN3 ($n = 3$). Lower panels show a zoom-in for regions indicated in full-size images. Mander's coefficients were determined to assess signal overlap between individual populations. Data are presented as mean values + SD ($n = 3$, cells examined over one experiment). Profile plots indicate the degree of colocalization for individual vesicles. Scale bars = 5 μM; Cntrl. control, IN input, WA wash, EL eluate. Source data are provided as a Source Data file.

When considering this shortlist, we identified >100-fold more cross-links compared to previously published whole-cell studies[28,29]. A probable key factor in this context was our starting material, consisting of SPIONs-enriched lysosomes, as lysosomal proteins are typically of low abundance relative to the total proteome (estimated 0.2% of cellular protein mass[30]). We also identified many cross-links from proteins that are seemingly unconnected to lysosomes. These proteins present possible substrates which were degraded in the lysosomal lumen during the time-point of enrichment, members of protein complexes associated with the lysosomal membrane, or unspecifically enriched proteins binding to the magnetic columns, which could be due to the large amount of cells utilized as input for lysosome enrichment (64 plates per replicate).

With respect to individual lysosomal proteins, we did not observe a strong correlation of their abundance and the number of annotated cross-links. We identified, for example, no cross-links for the lysosome-associated membrane glycoproteins 1/2 (LAMP1 and LAMP2), which were estimated to contribute 50% of lysosomal protein mass[69], and were among the most abundant lysosomal proteins in our dataset. Also, for the lysosomal luminal hydrolase CTSD, which was detected with the third-highest iBAQ value, only two unique cross-links were detected, while we found the same number for the low-abundant protein CHID1 (-275-fold less abundant). This could be due to several reasons: for the cross-linking of intact lysosomes, the lack of membrane permeability of reactive DSSO prevents the analysis of lysosomal luminal proteins or domains (in case of membrane proteins), and LAMP1/2 only contain a short cytosolic domain (-10 amino acids). While analysis of disrupted lysosomes should allow for coverage of luminal proteins/domains, as was, e.g., the case for PPT1, modification of amino acids by glycosylation could be a reason for the lack of identified cross-links for certain proteins, as the lysosomal luminal regions of LAMP1/2, as well as CTSD, are highly glycosylated. This could be due to interference of glycosylation with the cross-linking reaction, proteolytic digestion, or its influence on peptide fragmentation. Furthermore, in order to allow for the identification of post-translationally modified cross-linked peptides by database searching, the molecular weight of each modification needs to be defined. This further prevents the identification of glycosylated peptides if no special emphasis is put on their analysis.

Based on the identified cross-links, we generated a protein interaction network containing 70% potentially novel PPIs. Compared to other approaches, cross-linking allows for identifying interacting residues between individual proteins in situ. An approach able to generate data under similar near-native conditions is proximity biotinylation, utilizing, e.g., BirA* or APEX2, which have been used in two different studies to investigate the lysosomal (surface) proteome and interactome in vivo[15,70]. In contrast to cross-linking, proximity biotinylation only allows to determine the presence of a protein within a defined radius relative to the respective fusion construct. It cannot identify, however, if two proteins are directly interacting, or which residues/domains are involved. This is exemplified by a recent study utilizing BirA* fused to eight different proteins, including LAMP1, LAMP2, and LAMP3, for investigation of the lysosomal proteome[70]. Comparison of the interaction partners, which were identified for the individual LAMPs, revealed that only 8–18% were unique, while the others were identified for at least two of them. From these data, it can be inferred that the latter proteins are in close proximity to the lysosomal membrane, but it cannot be identified with which of the LAMP proteins they directly interact. This is not only the case for these three examples but for all lysosomal proteins which may be utilized as bait. If a specific cross-link is found for a given lysosomal protein, on the other hand, the respective amino acids have to be within a distance of 35 Å, implying direct interaction. Therefore, compared to proximity biotinylation approaches, the interactome presented in this study provides a level of detail that is unprecedented for the analysis of lysosomal PPIs. Furthermore, cross-linking allows for the unbiased analysis of PPIs while proximity biotinylation requires the expression of fusion proteins, possibly inducing artefacts due to changes in abundance, confirmation, or distribution relative to the endogenous population. Proximity biotinylation requires, however, less input material, and allows for the identification of PPIs irrespective of the presence of lysine residues at a suitable distance, facilitating the identification of a larger spectrum of interaction partners with higher sensitivity.

Another common approach for the investigation of protein interactions is co-IP, which we used to confirm two PPIs of members of the largest interaction networks, namely the FLOT1/FLOT2 complex and the V-ATPase complex. We selected the interaction of ATP6V1D with FZD9, a G-protein-coupled multi-pass transmembrane receptor for WNT2[71], as another member of this family, FZD8, was previously identified to interact with ATP6AP2[41]. Intriguingly, it was shown in this study that correct V-ATPase function, and accordingly acidification of the lysosome, is necessary for WNT signaling, and that interaction of FZD8 with the V-ATPase complex plays a decisive role[72]. The direct interaction of ATP6V1D and FZD9 identified in this study substantiates this functional connection of WNT signaling and lysosomal acidification, with a possible role related to the latter. V-ATPase can only acidify lysosomes when the V0 part, which is integrated in the lysosomal membrane, pairs with the cytosolic V1 part. The independently assembled V1 part[7] can reversibly dissociate from V0, which appears to be a process that can be regulated through different types of stimuli[39]. Similarly, it is conceivable that frizzled proteins could control V-ATPase V0/V1 assembly, and thus possibly regulate lysosomal acidification, and hence WNT signaling.

The second major lysosomal complex covered in our dataset was mTORC1. Among others, we detected cross-links for the interaction of the same lysine in the N-terminal region of LAMTOR1 with different residues of LAMTOR3. Both proteins are members of the Ragulator complex, whose interaction with the RAG GTPases regulates mTORC1 activity[73]. It was shown previously, that the N-terminal region of LAMTOR1 could not be crystallized without the RAG GTPases, implicating that it could exist in an unordered state under these circumstances[48]. Matching of the three LAMTOR1/3 cross-links to the Ragulator crystal structure determined in the presence of RAG GTPases fulfilled only for one of them DSSO's distant constraints. A possible explanation for the other two cross-links is that they originated from a state where Ragulator was not interacting with the RAG GTPases, indicating an alternative structure of the LAMTOR1 N-terminal region in situ.

When matching of cross-links to a structure determined from overexpression or in vitro experiments results in violation of DSSOs distance constraints, it can indicate that this structure does not present the native form of a protein or complex. This applied to the inter-links matched to the dimeric model of PPT1 (PDB identifier 3GRO), which were responsible for 43% of overlength cross-links in our dataset. On the other hand, all intralinks matched the published structure. This prompted us to further investigate possible routes of PPT1 assembly. Our proposed tetrameric model satisfied all distance constraints, which is in line with previous findings detecting the majority of enzymatic activity in size exclusion chromatography fractions correlating with the molecular weight of a tetrameric assembly[50]. The observed discrepancy to the crystallography-based structure could be related to the expression system utilized (*Spodoptera frugiperda*), as it lacks the capability to glycosylate PPT1 properly. In line with this possibility, it was shown previously, that glycosylation-deficient PPT1 variants are devoid of enzymatic activity, which was attributed to improper folding[50,74].

The flotillins, which yielded the highest number of cross-links, were reported in several studies to interact with a large variety of proteins at different subcellular locations and to be involved in a plethora of processes[75]. FLOT1 has been demonstrated to play a role in clathrin/caveolin-independent endocytosis[65], and the cargo of such FLOT1-positive endosomes has been shown to be delivered to lysosomes[63,76]. Furthermore, FLOT1 was detected at the lysosomes' cytosolic face[51,52]. In accordance with these findings, we detected and verified the interaction of FLOT1/FLOT2 with guanine-nucleotide-binding protein subunit beta-4 (GNB4), a beta subunit of heterotrimeric G-proteins[77]. Variants of GNB4 were shown to cause Charcot–Marie–Tooth disease[78], and downregulation of GNB4 levels has been found in Gaucher Disease, a lysosomal storage disorder[79]. Association of GNB4 with lysosomes was implicated in a previous study[80], and a yeast two-hybrid screen revealed its direct interaction with LAMP2[81], which co-localizes with FLOT1 and the lysosomal surface[51]. Taken together, these data provide further evidence that GNB4 is located at lysosomes, and imply that flotillins could play a role in its localization and/or function.

With respect to the structure of flotillins, only rudimentary experimental information exists based on NMR analyses of the mouse Flot2 N-terminal region (PDB identifier 1WIN). For the remaining protein, until recently, the predicted structure featured a 180° turn in the central helix of both flotillins resulting in direct interaction of the proteins' N- and C-terminal domains[43]. The FLOT1 and FLOT2 structures predicted by AlphaFold2 disagree with this model, featuring an extended helix with a bend in its middle for both of the proteins. This is supported by the cross-links detected in our dataset. We did not observe long-distance intralinks, which would be indicative of a major fold resulting in the proximity of distant regions of the proteins, and the inter-links of both proteins behave in unison, confirming intermolecular interactions along an extended region of the heterodimer.

We detected interaction with GNB4, among others, in the putative interaction hotspot, which localized in both flotillins to the major bend in the extended alpha-helical structure. Intriguingly, both the FLOT1 and the FLOT2 tyrosine residue located in this structure were reported in PhosphoSitePlus[82] to be phosphorylated in >30 studies. Therefore, they could present a potential regulatory element for PPIs, e.g., to discriminate interactions regulated by tyrosine kinase- or G-protein-coupled receptor signaling, as it was shown previously that flotillins are involved in both types of signaling pathways[83]. Deletion of this interaction hotspot in both FLOT1 and FLOT2 did not only abrogate interaction with GNB4, but also interfered with the formation of higher-order assemblies, providing further evidence for a prominent role in the formation of PPIs for this region of the flotillins.

Except for a few proteins[66], no substrates for flotillin-mediated endocytosis are known to date. Our analysis of its putative cargo identified >300 proteins, including a large number of membrane proteins, receptors, and associated proteins. We were able to confirm interaction of FLOT1 with eleven putative cargo proteins by co-IP, lending further credibility to our dataset. This included membrane-associated as well as transmembrane proteins. Among those, the latrophilins stood out, as all three members of this family of adhesion G protein-coupled receptors[84] were found to be significantly enriched. Furthermore, we were able to co-IP them with both flotillins and observed their colocalization with FLOT1/FLOT2-positive vesicular structures, raising the possibility that they may be targets for flotillin-mediated endocytosis. The fact that only a sub-population of latrophilins co-localized with flotillin-positive vesicles implies that they are either rapidly delivered for lysosomal degradation after their endocytosis, or that they are also endocytosed by other mechanisms, depending, e.g., on clathrin or caveolin.

## Methods
### Cell culture and enrichment of lysosomes
HEK293 (CRL-1573) and HeLa (CCL-2) cells were obtained from ATCC. Tissue culture plates (10 cm) were coated with 0.5 mg/mL poly-L-lysine (PLL) in 1× phosphate-buffered saline (PBS) for 20 min at 37 °C. On each plate, $6 \times 10^6$ HEK293 cells were seeded in full medium supplemented with 10% (v/v) superparamagnetic iron oxide nanoparticles (SPIONs) solution (DexoMAG40 from Liquid Research Limited, core size: 8 nm, hydrodynamic size: 50 nm, Fe content: 10 mg/ml, coating: dextran 40 kDa) and incubated for 24 h. Subsequently, cells were washed three times with pre-warmed PBS, fresh full medium was added, and cells were incubated for 24 h. Prior to harvesting, cells were washed twice with ice-cold PBS, scraped off the plates in 2 mL each of ice-cold isolation buffer (250 mM sucrose, 10 mM HEPES-NaOH pH 7.4, 1 mM CaCl₂, 1 mM MgCl₂, 1.5 mM MgAc, 1× cOmplete EDTA-free protease-inhibitor cocktail), and pooled. Cell suspensions from four plates each were dounced with 25 strokes in a 15 mL dounce homogenizer, and nuclei, as well as intact cells, were pelleted by centrifugation for 10 min at 600×*g*, 4 °C. The supernatant (post-nuclear supernatant, PNS) was transferred to a new tube, the pellet resuspended in 3 mL of isolation buffer, and dounced and centrifuged again. The supernatant from this step was combined with the first one, and the pooled PNS was used for lysosome enrichment using LS columns in combination with a QuadroMACS magnet (both from Miltenyi Biotec). Columns were equilibrated with 1 mL 0.5% (w/v) bovine serum albumin (BSA) in PBS, the combined PNS of two cell culture plates was applied to one column, and the flow through was collected. After three washing steps with 1 mL isolation buffer each, columns were removed from the magnet, and lysosomes were eluted twice in 1 mL of isolation buffer using a plunger. Individual eluate fractions were centrifuged for 30 min at 20,000×*g*, 4 °C, the supernatants were discarded, the pellets were resuspended in isolation buffer, and for each biological replicate (individual experiments from different passage numbers), the pellets from 64 plates were pooled. Protein concentrations were determined using the DC protein assay. The efficiency of lysosome enrichment and lysosomal integrity was assessed using the β-hexosaminidase assay[85]. Fractions obtained from lysosome enrichment (25 μL each) were combined with 8 μL of 10% Triton X-100 or 8 μL of PBS, followed by the addition of 50 μL reaction solution (100 mM sodium citrate pH 4.6, 0.2% (w/v) BSA, 10 mM para-nitrophenyl-N-acetyl-2-B-D-glucosaminide) in a 96-well plate format. Subsequently, the plate was incubated for 15 min at 37 °C, and 200 μL of stop solution (0.4 M glycine-HCl, pH 10.4) was added to the sample. Absorbance was measured at 405 nm on a microplate reader.

## Transfection of cells and enrichment of FLOT1-/FLOT2-positive early endosomes

In total, 64 tissue culture plates were coated with PLL (0.5 mg/mL), HEK293 cells were seeded at a density of $3.5 \times 10^6$ cells/plate, and cultivated as described above. After 24 h, cells were transfected with 6 µg of plasmid (1:1 mixture of FLOT1-FLAG and FLOT2-FLAG or GFP) using TurboFect. After 4 h, the cell culture medium was replaced, and cells were incubated for 48 h. In the first step, early endosome enrichment was performed in a similar way as lysosome enrichment using adjusted pulse-chase conditions. After addition of SPIONs solution to the cells (10% (v/v) final concentration), cells were incubated for 5 min at 37 °C, washed with pre-warmed PBS, and fresh full medium was added for a 5 min chase. Subsequently, the plates were placed on ice, cells were washed twice with ice-cold PBS, scraped off the plates in 2 mL each of ice-cold isolation buffer, and pooled. Cell suspensions from four plates each were pooled, dounced, and centrifuged for 10 min at 600×*g*, 4 °C. The PNS was transferred to a new tube, and after additional douncing of the pellet and centrifugation of the solution both PNS fractions were combined. Early endosomes were enriched using LS columns in combination with a QuadroMACS magnet. The eluate fractions of individual endosome enrichments were pooled, and FLOT1-FLAG/FLOT2-FLAG-positive endosomes were further enriched by magnetic anti-FLAG beads (80 µL). After the addition of beads, eluate fractions were incubated on an end-over-end rotator at 4 °C for 4 h, beads were separated from samples by magnetic force (DynaMag 2 magnet), and supernatants were transferred to new tubes. The beads were washed three times with 500 µL PBS, and FLOT1-FLAG/FLOT2-FLAG-positive endosomes eluted by incubation of the beads with 150 µL of 150 ng/µL 3× FLAG peptide in 1× TRIS (hydroxymethyl) aminomethane buffered saline (TBS, pH 7.6) for 30 min at 600×*g*, 4 °C in a thermomixer. Subsequently, beads were separated from samples by magnetic force and eluate fractions transferred to a new tube. Protein concentrations were determined using the DC protein assay. For the generation of FLOT1/FLOT2 deletion constructs (Δ 100), 100 amino acids between residue 200 and 300 were deleted by PCR using the Q5 Site-Directed Mutagenesis Kit (New England Biolabs). The following primers were used: FLOT1_FWD: AGACACCTTTTCCTGCTTGG, FLOT1_REV: GAGAAGTCCCAACTAATTATGCAGG, FLOT2_FWD: CTCCTTCTTGCACTCAGCTTCC, FLOT2_REV: GCCGAGGGTGAAAAGGTGAA. The successful deletion was verified by Sanger sequencing.

## Cross-linking of samples

For lysosome- and early endosome-enriched fractions, a portion of the eluate containing 500 µg and 200 µg protein, respectively, was transferred to a new tube. Intact organelles were pelleted by centrifugation for 30 min at 20,000×*g*, 4 °C, the supernatant discarded, and the pellets resuspended in isolation buffer at a protein concentration of 1 mg/mL. Lysosomal samples were cross-linked in two states (intact and disrupted) while endosomal samples were cross-linked only in the intact state. For disruption of lysosomes, resuspended samples were lyzed with a sonicator (Bioruptor Plus) at an amplitude of 40 with three cycles of 30 s each. All samples were cross-linked at final DSSO concentrations of 5, 2, 1, 0.5, 0.25 mM for investigation of the optimal cross-linker concentration (DSSO titration). For the XL-LC-MS/MS experiments of both lysosome- as well as endosome-enriched fractions, a final concentration of 0.25 mM DSSO was applied. After the addition of DSSO, the cross-link reaction was allowed to proceed for 30 min at room temperature, and quenched by the addition of TRIS-HCl pH 8.0 (20 mM final concentration). Subsequently, proteins were precipitated by the addition of acetone at a ratio of 4:1 (v/v) and incubation overnight at −20 °C. The next day, samples were centrifuged for 20 min at 20,000×*g*, 4 °C, the supernatant was discarded, the pellet was washed twice with ice-cold acetone, air-dried, and stored at −80 °C until further use.

## Sodium dodecyl sulfate-polyacrylamide gel electrophoresis (SDS-PAGE)

Polyacrylamide gels were prepared in-house. Both running and stacking gels were prepared with 10% (w/v) SDS, 40% (v/v) acrylamide, 10% (w/v) ammonium persulfate (APS), and 1% (v/v) tetramethylethylenediamine (TEMED), while 1.5 M TRIS-HCl pH 8.8 and 0.5 M TRIS-HCl pH 6.8 was used for running and stacking gels, respectively. Laemmli buffer[86] (4× stock, 240 mM TRIS-HCl pH 7.4, 4% (v/v) β-mercaptoethanol, 8% (w/v) SDS, 40% (w/v) glycerol, 4% (w/v) bromophenol blue) was added to samples (1× final concentration) followed by incubation for 10 min at 56 °C. Gel electrophoresis was performed at 80–140 V for up to 1.5 h. Gels were either stained overnight with Coomassie Brilliant Blue G-250 or with a Silver Stain Kit.

## Antibodies

The following primary antibodies were used in this study: goat anti LIMP2 (Cat# AF1966-SP,1:2000), mouse anti ACT2 (Cat# A5316, 1:4000), and rabbit anti LAMTOR1 (Cat # HPA002997, 1:1000) from Sigma-Aldrich; mouse anti CANX (Cat# 66903-1-AP, 1:20,000), mouse anti FLOT2 (Cat# 66881-1-Ig, 1:1500), mouse anti FZD9 (Cat# 67023-1-Ig, 1:1500), rabbit anti ATP6V1D (Cat# 14920-1-AP, 1:1000), rabbit anti CTSD (Cat# # 21327-1-AP, 1:1000), rabbit anti FLOT1 (Cat# 15571-1-AP, 1:2000), rabbit anti GNB4 (Cat# 11978-2-AP, 1:2000), and rabbit anti SDHA (Cat# 14865-1-AP, 1:800) from Proteintech; goat anti LIMP2 (Cat# AF1966-SP, 1:2000) from R&D system; mouse anti ATP6V1B2 (Cat# SC166045, 1:1000) from Santa Cruz; mouse anti FLOT1 (Cat# 610821, 1:200), mouse anti FLOT2 (Cat# 610383, 1:200), and mouse anti GM130 (Cat# 610822) from BD Biosciences; mouse anti GAPDH (Cat# 5174, 1:2500), rabbit anti EEA1 (Cat# 2411, 1:200), rabbit anti FLOT1 (Cat# 18634, 1:200), and rabbit anti RRAGA (Cat# 4357, 1:1000) from Cell signaling; rabbit anti LPHN2 (Cat# NBP2-58704, 1:100) and rabbit anti LPHN3 (Cat# NLS1138, 1:200) from Novus Biologicals; goat anti-mouse IgG (H + L)-Alexa Fluor 488 (Cat# A-11029, 1:400), rabbit anti ATP6V1A1 (Cat# PA5-29191,1:2000), and rabbit anti LPHN1 (Cat # PA5-77475, 1:200) from Thermo Fisher Scientific; rabbit anti TUBA (Cat# 600-401-880, 1:2000) from Rockland; mouse anti LAMP2 (Cat# H4B4, 1:1000 for WB, 1:100 for IF) from Hybridoma Bank. rabbit anti DSSO (Self-made[35], 1:5000). The following secondary antibodies were used in this study: rat anti-FLAG (Cat# SAB4200119,1:10,000) from Sigma-Aldrich; HRP-coupled goat anti-mouse IgG (Cat# 115-035-044, 1:5000), Cy3-coupled goat anti rabbit IgG (H + L) (Cat# 111-165-144, 1:400), and HRP-coupled goat anti rabbit IgG (Cat# 111-035-003, 1:5000) from Dianova.

## Immunoprecipitation of proteins

HEK293 cells were seeded at a density of $6 \times 10^6$ cells/10-cm plate and cultivated for 48 h in full medium. Cells were washed twice with ice-cold PBS, scraped off the plate in 1 mL of ice-cold PBS, transferred to a microtube, and centrifuged for 10 min at 600×*g*, 4 °C. The supernatant was discarded and the pellet resuspended in either 300 µL of RIPA lysis buffer (50 mM TRIS-HCl pH 7.4, 150 mM NaCl, 1% Triton X-100, 0.1% (w/v) SDS, 0.5% (w/v) sodium deoxycholate (SDC), 1× cOmplete EDTA-free protease-inhibitor cocktail, 1 mM EDTA) or 300 µL of lysis buffer (50 mM TRIS-HCl pH 7.4, 150 mM NaCl, 1% NP-40, 1× cOmplete EDTA-free protease-inhibitor cocktail, 1 mM EDTA). Samples were incubated on ice for 30 min and passed through a 25 Gauge needle every 10 min. Subsequently, the lysate was cleared by centrifugation for 15 min at 20,000×*g*, 4 °C, transferred to a new pre-cooled microtube, and protein concentrations were determined using the DC protein assay. For each sample, lysate containing 1.6 mg of protein was incubated with 3 µg of antibody overnight by end-over-end rotation at 4 °C. The next morning, 60 µL of Protein A beads were added to each sample, followed by end-over-end incubation for 1 h at 4 °C. Beads were pelleted by centrifugation for 5 min at 1000×*g*, 4 °C, supernatants transferred

to new tubes, and beads were washed three times with 500 μL of ice-cold PBS. Proteins were eluted from the beads by incubation in 2× Laemmli buffer for 30 min at 45 °C.

## Blue native polyacrylamide gel electrophoresis (BN-PAGE)

Samples were supplemented with solubilization buffer (10 mM HEPES-NaOH pH 7.4, 1% (v/v) digitonin, 2 mM EDTA, 50 mM NaCl, 10% (v/v) glycerol, 1 mM phenylmethylsulfonyl fluoride (PMSF)) and loading dye (10 mM Bis-TRIS-HCl pH 7.0, 0.5% (w/v) Coomassie Brilliant Blue G-250, 50 mM ε-amino n-caproic acid), and loaded to self-cast blue native gradient gels (4–12% (w/v) acrylamide, 0.19–0.40% (w/v) bis-acrylamide, 67 mM ε-amino n-caproic acid, 50 mM Bis-Tris/HCl pH 7.0). The anode buffer was 50 mM Bis-TRIS-HCL pH 7.0, and the cathode buffer 50 mM Tricine pH 7.0, 15 mM Bis-TRIS-HCl pH 7.0, 0.2 % (w/v) Coomassie Brilliant Blue G-250. The temperature of the gel chamber was maintained at 4 °C and electrophoresis was performed at 50 V for 20 h.

## Western blotting

Proteins were transferred to nitrocellulose or polyvinylidene fluoride (PVDF) membranes using a semi-dry or wet electro blotter for 1 h or 2 h at 200 mA/membrane. Membranes were blocked in 5% nonfat dry milk in TBS containing 0.05% (v/v) Tween 20 (TBS-T) for 1 h at RT followed by incubation with primary antibodies overnight at 4 °C. The next day, membranes were washed three times with TBS-T for 10 min at RT followed by incubation with secondary antibody for 60 min at RT. Subsequently, membranes were washed three times for 10 min at RT with TBS-T, and the blots were developed. Protein expression signals were detected using the enhanced chemiluminescence (ECL) kit, visualized with the FUSION SOLO 4 M system, and analyzed by the FusionCapt advance software. Uncropped and unprocessed scans can be found in the Source Data File in the Supplementary Information. Individual images present a merged image of the marker and the respective chemiluminescence protein signals.

## Immunofluorescence microscopy and image analysis

HeLa and HEK293 cells were seeded in 12-well plates at a density of $2 \times 10^4$ and $4 \times 10^5$ cells per well, respectively. For HEK293 cells, glass coverslips were coated using PLL (0.5 mg/mL). Cells were cultured for 36 h after seeding and transfected with 1 μg of plasmid (1:1 mixture of FLOT1-GFP/FLOT2-FLAG and FLOT2-mCherry/FLOT1-FLAG) using TurboFect. After 4 h, fresh medium was supplemented and cells were incubated for 48 h. For staining, cells were washed using PBS and fixed with ice-cold methanol at −20 °C for 20 min. After fixation, cells were washed twice with PBS and blocked with 2% BSA in PBS for 1 h at RT. Blocked cells were stained with primary antibodies overnight at 4 °C in a humid chamber. Subsequently, cells were washed three times with TBS for 5 min each and incubated with secondary antibodies for 1 h at RT in the dark. Coverslips were washed three times with TBS for 5 min each, rinsed once with distilled water, and mounted on specimen slides using ROTI Mount FluorCare DAPI. Images were captured using Leica SP5 AOBS with SMD confocal microscope equipped with an HCX PL APO ×63/oil objective and 2× single-photon avalanche diode detector. Images were acquired using the LAS AF software. Images were annotated using Fiji software. Colocalization analyses and determination of Mander's coefficients (MCOE) were performed with the JACoP plugin[87].

## Mass spectrometry sample preparation

Precipitated proteins were resuspended in 100 μL of freshly prepared 8 M urea/100 mM triethylammonium bicarbonate (TEAB) and incubated for 45 min at 600 rpm, 37 °C. Disulfide bridges were reduced with 4 mM DTT (final concentration) at 56 °C for 30 min, alkylated with 8 mM chloroacetamide (final concentration) at RT for 30 min[88], and the reaction was quenched by the addition of 4 mM DTT. Subsequently, samples were diluted 1:1 with 100 mM TEAB, rLysC was added at an enzyme-to-protein ratio of 1:100 (w/w), and proteolytic digestion

was performed at 37 °C overnight. The following day, the urea concentration was reduced to 1.6 M by the addition of 100 mM TEAB, trypsin was added at an enzyme-to-protein ratio of 1:100 (w/w), and the samples were incubated at 37 °C for 8 h. The resulting peptides were desalted using 50 mg Sep-Pak $C_{18}$ cartridges, dried using a vacuum centrifuge, and stored at −80 °C until further use.

## Strong cation-exchange (SCX) chromatography fractionation

SCX fractionation was performed with an UltiMate 3000 RSLC HPLC chromatography system in combination with a PolySULFOETHYL A column (150 mm × 1 mm, 5-μm particle size). Desalted peptides (500 μg each) were reconstituted in 20 μL of SCX solvent A (20% acetonitrile (ACN), 10 mM $KH_2PO_4$ pH 2.7), loaded to the analytical column with 100% SCX solvent A, and eluted with increasing amounts of SCX solvent B (500 mM KCl, 20% ACN, 10 mM $KH_2PO_4$ pH 2.7) at a flow rate of 50 μL/min. The gradient was as follows (adapted from[38]) 0–42 min: 0–2% B; 42–50 min: 2–3% B; 50–60 min: 3–8% B; 60–70 min: 8–20% B; 70–80 min: 20–40% B; 80–86 min: 40–90% B; 86–90 min: 90% B; 90–91 min: 0% B; 91–120 min: 0% B. Eluting peptides were collected with a fraction collector, individual fractions dried using a vacuum centrifuge, resuspended in 20 μL of 5% ACN, 5% formic acid (FA), and desalted using $C_{18}$ STAGE-tips[89]. STAGE tip eluates were dried using a vacuum centrifuge, and resuspended in 5% ACN, 5% FA, and peptide concentrations were determined with a quantitative fluorometric peptide assay.

## Liquid chromatography-electrospray ionization tandem mass spectrometry (LC-ESI-MS/MS)

Dried peptides were resuspended in 5% ACN, 5% FA and analyzed using an UltiMate 3000 RSLCnano UHPLC system coupled to an Orbitrap Fusion Lumos mass spectrometer (Data acquisition with Orbitrap Fusion Lumos Tune Application (3.4) and Xcalibur (4.2)). From each sample (25% of the total amount for individual SCX fractions, 1 μg of non-cross-linked samples) were loaded on a 50 cm $C_{18}$ reversed-phase analytical column at a flow rate of 600 nL/min using 100% solvent A (0.1% FA in water). For analytical column preparation, fused silica capillaries (360 μm outer diameter, 100-μm inner diameter) were used to generate spray tips using a P-2000 laser puller. Tips were packed with 1.9 μm ReproSil-Pur AQ $C_{18}$ particles to a length of 50 cm. Peptide separation was performed with 120 min (SCX fractions) and 240 min (non-cross-linked samples) linear gradients from 5–35% solvent B (90 % ACN, 0.1% FA) at a flow rate of 300 nL/min. MS1 spectra were acquired in the Orbitrap mass analyzer from $m/z$ 375 to 1575 at a resolution of 60,000. For peptide fragmentation, charge states from 3+ to 8+ for cross-linked samples and 2+ to 5+ for non-cross-linked samples were selected, and dynamic exclusion was defined as 60 sec and 120 sec for 120 min and 240 min gradients, respectively. Cross-linked samples were either analyzed with an MS2-MS3-MS2 strategy[90] or with a stepped collision energy approach[91], where ions with the highest charge state were prioritized for fragmentation. For both methods, MS2 scans were acquired at a resolution of 30,000 in the Orbitrap analyzer with a dynamic mass window. In case of stepped collision energy, peptides were fragmented using higher collision dissociation (HCD) with 21, 26, and 31% normalized collision energy (NCE). For the MS2-MS3-MS2 fragmentation method, sequential collision-induced dissociation (CID) and electron-transfer dissociation (ETD) spectra were acquired for each precursor. The precursor isolation width was set to $m/z$ 1.6 with standard automatic gain control and automatic maximum injection time. The NCE for CID-MS2 scans were set to 25%, and calibrated charge-dependent ETD parameters enabled. MS3 scans were triggered by a targeted mass difference of 31.9721 detected in the MS2 scan. The MS3 scan was performed in the ion trap part of the instrument with CID at 35% NCE with normalized automatic gain control (AGC) target of 300%. DIA analyses were performed with the same instrumental setup as described above. For each sample, 1 μg of peptides were loaded

directly on a reversed-phase analytical column packed with 3-μm Reprosil-Pur AQ C$_{18}$ particles to a length of 40 cm and separated with 120 min linear gradients. After the acquisition of one MS1 scan 24 static windows, DIA MS2 scans were performed. MS1 scans were acquired in the Orbitrap analyzer from $m/z$ 350 to 1200 at a resolution of 120,000 with a maximum injection time of 20 ms and an AGC target setting of $5 \times 10^5$. MS2 scans were defined to cover the MS1 scan range with 36 scan windows of 24.1 $m/z$ each, resulting in an overlap of 0.5 m/z and a cycle time of 3.44 s. Peptides were fragmented by HCD with an NCE of 27%, and spectra were acquired in the Orbitrap analyzer with a resolution of 30,000, a maximum injection time of 60 msec, and an AGC target setting of $1 \times 10^6$.

## Proteome discoverer analysis
Thermo *.raw files from cross-linked samples were analyzed using Proteome Discoverer 2.4, utilizing Mascot 2.5.1 and the XlinkX node for peptide identification. The following settings were used for both algorithms: precursor ion mass tolerance: 10 ppm; Orbitrap fragment ion mass tolerance: 20 ppm; ion trap fragment ion mass tolerance: 0.5 Da; fixed modification: carbamidomethylation at cysteine; variable modification: oxidation at methionine; enzyme: trypsin; the number of allowed missed cleavage sites: 2; minimum peptide length: five amino acids; cross-linking site: lysine (K) and N-terminus of proteins. Data were searched against UniProt *Homo sapiens* (entries: 20,365, release date: 05/2020) in combination with the common repository of adventitious proteins (cRAP) database containing common contaminants. The Proteome Discoverer workflow was split into two branches with a cross-link and standard peptide search. MS2 spectra containing DSSO reporter ions were analyzed with pre-defined "MS2-MS2-MS3" and "MS2" search options using XlinkX. Peptide identifications were accepted with a minimum XlinkX score of 40 and filtered at FDRs of 1 and 5% at the cross-linked peptide spectrum level. Cross-links were exported. Spectra, which did not contain reporter ions were searched using Mascot. Identified peptides were filtered at 1% FDR on the peptide level using Percolator and proteins exported at 1% FDR. Only high-confidence peptide identifications were considered and data exported. Data from both algorithms were further analyzed applying different software packages (R, Excel 2019, GraphPad Prism 6.01, STRING 11.0, Cytoscape 3.8.0, xiVIEW (07/2019), TopoLink (06/2019), PSIPRED 4.0, PCOILS (07/2018), and PyMol 2.3).

## MaxQuant analysis
Thermo *.raw files from non-cross-linked samples were analyzed using MaxQuant 2.0.3[92] for determining iBAQ values[37]. The following settings were used: precursor ion mass tolerance: 4.5 ppm; Orbitrap fragment ion mass tolerance: 20 ppm; fixed modification: carbamidomethylation at cysteine; variable modifications: oxidation at methionine, acetylation at protein N-terminus, and deamidation at asparagine (N) as well as glutamine (Q); enzyme: trypsin; the number of allowed missed cleavage sites: 2; minimum peptide length: 5 amino acids. Data were searched against UniProt *Homo sapiens* (Entries: 20,365) in combination with the cRAP database containing common contaminants. Data were filtered at 1% FDR on the peptide level and protein level and exported, followed by analysis with different software packages (Excel and GraphPad Prism).

## Spectronaut analysis
Thermo *.raw DIA files from FLOT1-FLAG + FLOT2-FLAG-transfected HEK293 cells were analyzed using Spectronaut 14.7.20. Initially, hybrid spectral libraries were generated from both DDA and DIA files with the Pulsar search engine integrated into Spectronaut, applying the following parameters: precursor ion mass tolerance: dynamic; Orbitrap fragment ion mass tolerance: dynamic; fixed modification: carbamidomethylation at cysteine; variable modifications: oxidation at methionine, acetylation at protein N-terminus and deamidation at

asparagine (N) as well as glutamine (Q); enzyme: trypsin; the number of allowed missed cleavage sites: 2; minimum peptide length: five amino acids. Data were searched against UniProt *Homo sapiens* (entries: 20,365, release date: 05/2020) in combination with the cRAP database containing common contaminants. For each peptide, the 3−6 most abundant b/y ions were selected for library generation, dependent on their signal intensity. Dynamic retention time alignment was performed based on the high-precision indexed retention time (iRT) concept[93]. Mass tolerances (precursor and fragment ions), as well as peak extraction windows, were defined automatically by Spectronaut. Normalization was disabled, and data filtered at 1% FDR on the peptide and protein level (*q*-value <0.01). High-confidence identifications were exported, followed by analysis with different software packages (R, Excel, STRING, Cytoscape, and GraphPad Prism).

## Structural analysis
Protein sequences were obtained from UniProt and searched using BLAST 2.9.0 against protein database (PDB) entries with an *E*-value of 0.0001. In case no reference structure for *Homo sapiens* was available, structures from other organisms were obtained from the SWISS-MODEL repository (09/2021)[46], and/or predicted structures from the AlphaFold2 protein structure database were used[47]. Amino acid numbering was adjusted to UniProt entries, identified cross-links were mapped and topologically evaluated with TopoLink (06/2019)[45], and visualized using PyMol 2.3.

## Molecular docking
Protein structure perturbation and optimization was performed with SCWRL 4.0[94] and restraint-based docking with HADDOCK2.2[95] as well as CNS 1.3[96]. Distance constraints of identified cross-links (20 ± 10 Å) were used to limit the possible interaction search space, applying unambiguous restraint distances on C-beta (except for glycine, for which C-alpha was used). In line with the default HADDOCK protocol, 500 initial restraints-based complex models were generated, followed by their rigid-body energy minimization. For the best 100 models, semi-flexible refinement in torsion angle space was performed, followed by molecular dynamics refinement in explicit water. Generated models were evaluated based on the weighted sum of electrostatic and van der Waals energies, complemented by the empirical desolvation energy. Based on these parameters, the ten best-scoring models were reported. Finally, models were further clustered within a 5 Å pairwise root mean square deviation, and the lowest energy model of each cluster reported. Results were visualized using PyMol 2.3.

## Identification of putative FLOT-endosome cargo by statistical analysis
Spectronaut results were analyzed using R (R Core Team, 2020) applying the integrated development environment RStudio 1.3.1056. For data importing, tidying, transforming and statistical modeling Tidyverse 1.3.0[97] and R base packages were used. Results were exported using Openxlsx 4.1.5 and visualized by Ggplot 3.3.2 and Viridis 0.5.1. Protein signal intensities were initially log2-transformed for data quality assessment and visualization. Missing values were replaced by "NA", while imputation of missing values was omitted. Subsequently, data from individual replicates of the experimental conditions (GFP, SPIONs, and SPIONs+IP) were categorized into three populations as follows: background (three valid values in all three conditions), SPIONs-specific (no valid GFP value and ≥2 valid values in SPIONs or SPIONs+IP) and SPIONs "on/off" (≥2 valid values in only one SPIONs condition and no valid value in other conditions). Subsequently, non-logarithmic data of the individual datasets were normalized on the signal intensities of FLOT1 and FLOT2 in the respective datasets, while the GFP sample was not normalized, followed by log2 transformation of all datasets. Proteins with ≥2 valid values in each dataset were compared using a two-sided unpaired *t* test. On/off proteins were

defined as significant and their *P* values set to zero; while *P* values of proteins not matching any of the two conditions were set to one. *P* value adjustment was performed according to Benjamini–Hochberg[98] and proteins with a *q*-value ≤0.05 were considered differently enriched. Significantly enriched proteins were submitted to protein–protein interaction and enrichment analysis with STRING 11.0[99] while the entire list of proteins found in the experiment was defined as a background gene set. Networks (type: full) were generated, in which the edges indicate both functional and physical protein associations.

### Reporting summary

Further information on research design is available in the Nature Research Reporting Summary linked to this article.

## Data availability

The raw MS-data data generated in this study have been deposited to the ProteomeXchange Consortium via the PRIDE partner repository database and are publicly available as of the date of publication under the accession code PXD030532 (XL/DIA-LC-MSMS and LC-MSMS data of lysosome-/early endosome-enriched fractions). The integrative cross-link models for PPT1 and FLOT1/FLOT2, as well as all docking input/output parameters generated in this study, are provided in Supplementary Data 8 and 9. Protein structure data used in this study are available from the PDB database. List of PDB accessions: 6WLW (ATP6V0B), 6WLZ (ATP6V1A), 6B9X (LAMTOR3), 5Y39 (LAMTOR5), 5TDH (GNAI1), 5FWK (HSP90AB1), 5ULS (HSP90B1), 3WGD (TXNDC5), 3TJB (PRDX4), 4XCS (PRDX1), 3GRO (PPT1), 1W7B (ANXA2), 1R46 (GLA), 3WEZ (GLB1), 6CES (NPRL3), 3PDF (CTSC), 5UX4 (CTSD), 1GL2 (STX7), 6ASY (HSPA5), 3C7N[] (SSE1), 6WM2 (ATP6V1E1), 5Y3A (LAMTOR1-5), 5H64 (MTOR, RPTOR, MLST8), 5HE0 (GNB1, GNG2, GRK2), 3LMY (HEXB), 2GJX (HEXA), 6JMQ (SLC7A5). Source data are provided with this paper.

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

## Acknowledgements

The authors would like to thank Stephane Bodin and Cecile Gauthier for sharing of plasmids containing GFP/RFP-tagged FLOT1/FLOT2 as well as Norbert Rösel and Nadine Weiberg for help with experiments. J.S. received funding by the Studienstiftung des Deutschen Volkes, P.M. by the Jürgen Manchot Stiftung, and H.E. by the IMPRS "From Molecules to Organisms". M.S. and M.E. were funded by the de.NBI project (FKZ 031A 534A) of the Bundesministerium für Bildung und Forschung (BMBF) and D.W. received funding from the FOR2625 of the DFG (Deutsche Forschungsgemeinschaft, German Research Foundation). We would like to thank the Microscopy Core Facility of the Medical Faculty at the University of Bonn for providing help, services, and devices funded by the DFG project number 169331223.

## Author contributions

Conceptualization: J.S. and D.W.; methodology: J.S., V.G., D.W.; bioinformatic analysis and structural modeling: J.S., H.E., M.S., and O.K.; investigation: J.S. and P.M.; writing—original draft: J.S. and D.W.; writing—review & editing: J.S., H.E., P.M., M.E., O.K., V.G., and D.W.; visualization: J.S., H.E., and P.M.; funding acquisition: J.S. and D.W.; resources: V.G. and D.W.; supervision: D.W. All authors contributed to the final manuscript.

## Funding

## Competing interests

The authors declare no competing interests.
