## [Peer Review File · Nature Communications]

REVIEWER COMMENTS

Reviewer #1 (Remarks to the Author):

The authors have used crosslinking and mass spectrometry on endosomes and lysosomes isolated from human cells to identify protein complexes of these organelles. The follow up with characterization of several of the interactions that they identified as a validation of these results. This includes a PPT1 tetrameric complex; FZD9 interactions with a vATPase subunit; and flotillin interaction with GNB4. They furthermore present data in support of heterodimers of flotillins that may assemble into large multimers to support cargo trafficking and propose latrophilins as a putative cargo of flotillin-dependent trafficking. The experiments are generally performed with a high degree of rigor and interpretation of the results is reasonable. This data should be of a useful resource for investigators of lysosome cell biology. The new insights regarding the proteins mentioned above are intriguing but have not been developed with sufficient depth to confidently demonstrate biological relevance. However, I recognize that fully defining such relevance is beyond the scope of this manuscript. As a result, I see greatest value as a resource for investigators in the lysosome cell biology community but am less confident about the extent to which each individual finding will stand on its own. I therefore have concerns that the title promises more than the article delivers. I also have some concerns about specific issues that I have summarized below.

1. Figure 5 A and B are meant to show partial localization of flotillins to lysosomes. However, the magnification of the images is not adequate to appreciate this message and it is not clear whether this represents flotillin localization to subdomains on lysosomes or just coincidental (and possibly non-specific) overlap. It is furthermore not clear if the LAMP2 signal is specific as it does not clearly identify individual lysosomes but is found throughout the cytoplasm and is unexpectedly enriched in the nucleus.

2. Colocalization of flotillin and latrophilin in Figure 7B is also not well developed. It is not clear how much of each signal is specific; how much overlap is coincidental and whether this is occurring on endosomes/lysosomes.

3. Kaushik references not defined on page 12.

Reviewer #2 (Remarks to the Author):

The study of Jasjot Singh and collaborators entitled « Cross-linking of the Endolysosomal System Reveals Flotillin Structures and Putative Cargo » aims to identify the proteins of the lysosomes in HEK293T cells and their interactions. They use the technique of cross-linking mass spectrometry on lysosome enriched fractions, kept intact or permeabilized to favor the penetration of the cross-linking agent. From 2467 unique residue-to-residue cross-links, they identified 270 intra- and 254 inter-protein interactions. The highest numbers of PPI were identified for the V-ATPase, the flotillins, mTORC1 and the syntaxins.

Some global and more specific data of protein-protein interactions from these lysosome enriched fractions (for the V-ATPase and the flotillins) are presented in figure 2. Then they combined the cross-link data with available data from cristalography or generated using AlphaFold, a new tool recently available to propose the strucutre and assembly of a panel of proteins (alpha-galactosidase A, V-ATPase subunits, and PTP1 dimer and tetramer). Then the authors focused on flotillins, for which they identified the highest number of cross-links. Using a combination of several bioinformatic tools that predcte secondary structure, 3D structures combined with the cross-link data, and the integration of known structural features, they propose a model for the flotillin 1/flotillin 2 homodimer.

Then they also use the technique of cross-linking mass spectrometry on early endosomes enriched fractions from cell that were co-transfected with Flag-tagged FLOT1 and FLOT2. They identified 1081 cross-links from 414 unique proteins. The analysis of the FLOT1-FLOT2 cross-links revealed that they are almost similar (80%) to those from lysosome-enriched fractions. Then the authors proposed a model for the oligomerization of flotillins. Finally, they performed proteomic analysis using the early endosome enriched fraction by SPIONs and using immunoprecipitation against Flag-tagged flotillins in order to propose an identification of the flotillin cargos in Hek293 cells.

The data obtained are potentially interesting but raised several questions, in particular why so many proteins identified are non-lysosomal proteins ? why abundant proteins of the lysosome where not found engaged in PPI ? Moreover a number of specific points need to be addressed (see below).

The organization of the article is a bit confusing and the presentation of the different examples is a bit of a catalog. The authors should think about to reorganize the data in figure 2, 3, 4

Remarks

1) This study is perform in Hek293T cells. Why having chose these cells ?

Will the proteins identified in the lysosomal fraction vary depending on the cells chosen? Would the proteins that have been studied more precisely (flotillins) also be present in other cell types?

2) Preparation method. As the SPIONs technique is the basis for characterization studies of lysosome or early endosome proteins, it is important that all elements to understand the experimental procedures are given.

A better description of the ethod sholud be given. What is the size, nature of the beads ?

The bibliographic reference that is mentioned is not available.

Line 663 : DexoMAG40 is unavailable. Is it the same as SphereMAG ? Please be sure to provide the details.

Line 672 : define what is a LS column.

Line 252 : the authors said that the length of the DSSA lonk is around 35A. In the technical sheet (<https://www.thermofisher.com/order/catalog/product/A33545>) it is mentioned 10,5A. Why this difference ?

It will be important to show the accumulation of the beads in lysosomes by electron microscopy ? or using fluorescent microscopy technique ?

From the western blots showed in figure 1C, through the quantification of the lysosomal proteins in the whole cell lysate and in the lysosomal enriched fraction, it should be possible to estimate the enrichment of lysosomal proteins in the preparation.

3) Table S3 and Figure 2 :

The authors claim that they generated a protein-protein interaction network containing 70% potential novel protein-protein interactions. But what is puzzeling is that the analysis of the interlinks led to the identification of many proteins that are not lysosomal proteins (table S3). Are these proteins false-positive ? Or are these proteins there because they undergo a degradation process ? Or are they lysosomal protein not identified yet?

The fact that many proteins identified by the PPI are non lysosomal protein has to be discussed in more details.

4) Equally confusing is the fact that abundant lysosome protein families are not identified, such as LAMPs or tetraspanins. This is just mentioned in the discussion but no explanation is provided. How do you explain this result?

Considering this point the title (line 200) « Characterization of the Human Lysosomal Interactome » is a bit of a stretch.

Also, in the discussion, the authors mention a study using BirA to identify LAMP partners (Go et al, Nature 2021). Although few partners where identified in this overall study, it is more than zero, as was the case in the submitted manuscript.

Lines 529-532, what do the authors mean? Not clear. What will be more interesting is to compare the advantages and disadvantages of each technique.

5) Line 207 the authors say that lysosomal PPI « revealed an overrepresentation of nuclear and cytoplasmic/cytoskeletal proteins ». Are these the same proteins identified in the study by Go et al, Nature 2021 considering the lysosomal proteins associated partners ?

6) Figure 2H and I

Except one case of PPI between Flot2 with another protein (around aa 250), in all cases the PPIs between FLOT1-FLOT2 are located at the same aa sequence than the FLOT1/2-other

How the same region could be involved in FLOT1/FLOT2 interaction and FLOT1/2-other proteins interactions ?

Lanes 227-229 : not clear, to better explain.

7) Why the authors splitted the data on the V-ATPase at two distinct parts of the results (lines 210-218 and 259-272) ?

Idem concerning the flotillin data. Why having split them between figure 2 and 4 ?

8) Line 325 and line 584: the authors mentioned a Flot2 Nterm NMR structure. No reference is provided.

Reference #54 does not present such data.

9) Lines 340-342 : The authors selected model four among the 5 models for the FLOT1-FLOT2 heterodimer obtained using ColabFold. This is not clear why ? Couldn't these different models co-exist ? The added value of the analysis performed using cross-linking and proteomics is not emphasized.

These softwares alone are already a revolution for obtaining 3D structures of proteins and their complexes. What is the difference between the FLOT1-FLOT2 heterodimer obtained by ColabFold alone compared to the one obtained by combining ColabFold and the cross-link data ?

10) Could it be possible to analyze the impact of the mutation of the tyrosine phosphorylation sites, which were shown to be crucial for flotillin-mediated endocytosis and FLOT1/FLOT2 interaction (Y160 in FLOT1 and Y163 in FLOT2), on the FLOT1-FLOT2 heterodimer obtained by ColabFold ?

11) Immunostaining for flotillins.

Flotillins are well known to be located in lysosomes/late endosomes. The images presented in figures 5 and 7, S6 and S7 are wired. In HeLa cells these stainings do not reveal vesicular structures such as lysosomes/late endosomes. The stainings reveal filamentous structures but not vesicles.

Being able to see colocalizations with these types of images is not the guaranty of the real colocalization in structures such as LE.

It will be important to try other fixation/permeabilisation methods and to really identify flotillins upon expression of fluorescently-tagged versions.

Also comparing the images of Figure 5A to those of Figure S5A for HeLa cells in particular, the distribution of the FLOT/FLAG (Fig S5A) does not match with the immunostaining (Fig 5A).

The quantification of the colocalization is required.

The inset are very small and have to be enlarged. Showing so many cells is not important, it will be helpful to show only 2/3 enlarged cells and to associate a colocalization analysis.

12) In the reference mentioned line 581 (Kauchik), it was shown electron microscopy images showing for example the co-localization of LAMP2 and flotillin1. This makes again puzzling the absence of LAMP2/FLOT1 cross-links. The authors said (lines 581-583) « Taken together, these data provide strong evidence that the lysosomal localization of GNB4 is mediated through interaction with the flotillins. » At the same time they said that a yeast two-hybrid screen revealed a direct interaction of GNB4 with LAMP2. Why flotillins should be involved then ?

13) Why having analyzed proteins in EE ?

14) How the model in figure 5I was obtained ? No indication neither in the text nor the figure legend are mentioned.

15) How the author try to model the tetrameric FLOT1-FLOT2 complex ? They need to explain that and to discuss on which basis this model was proposed.

Line 602 they talk about « an abundant tetrameric assembly », this appears contradictory ?

16) What is the impact of mutation of the Y residues proposed to be involved in FLOT1/FLOT2 interaction on the model figure 5I ?

17) Why the authors said « A potential function for such an assembly could be the formation or transport of flotillin-positive endosomes » ? This has to be explained.

18) Amongst the 300 proteins identified by co-IP with flotillins and proposed as flotillin cargos, how many are found in the lysosome-enriched fraction ?

19) From table S7 it is not obvious that many proteins are membrane or transmembrane proteins as shown in figure S6F. In this table many proteins are RNA-binding proteins, DNA-binding proteins. Could these proteins be flotillin cargo ? Why they are not present in figure 6C ?

20) Concerning the discrimination of « protein populations that were enriched/depleted in FLOT1-/FLOT2-positive endosomes relative to the total cellular pool of early endosomes », the experimental procedure should be better explain. Where the data of the proteins depleted in the FLOT1-/FLOT2-positive endosomes are found.

Minor points

- Lines 575 and 581 : the references were not included

- Line 854 : what is PDB : 1 WIN ?

Reviewer #3 (Remarks to the Author):

In their study Singh et al. investigate lysosomes and early endosomes by MS-based proteomics. In a major experimental effort, the authors use multiple MS-based proteomic workflows, namely cross-linking mass spectrometry (XL-LC-MS/MS), to identify the protein content and protein protein interactions (PPIs) from enriched lysosomes and early endosomes.

In a first part the authors use superparamagnetic iron oxide nanoparticles (SPIONS) to enrich lysosomes followed by cross-linking of intact and disrupted lysosomes. By comparison with known lysosomal proteins, the authors can convincingly show that they are able to enrich lysosomes and by applying an established cross-linking workflow - which the authors apply the first time to lysosomes -

they are able to detect cross-links on multiple proteins, including in particular the V-ATPase and flotillins. Here, the authors identify potentially novel interactors – namely FZD9 for the ATPase and GNB4 for FLOT1 and 2 – which the authors subsequently validate by Co-IP. The authors then move on to use the identified cross-links for structural refinement of the V-ATPase and the PPT1 complex, which the authors convincingly show to likely exist as a tetramer within lysosomes. This part of the work is very solid and clearly increases our knowledge about lysosomes.

In a next step the authors then use their cross-linking data and a full-length structural monomeric model from AlphaFold to generate a dimeric model of the FLOT1/FLOT2 dimer using HADDOCK (FLOT1/FLOT2 was known previously to form a dimer). The authors then move on to make precise predictions based on this model. However, the model suggests a level of detail that is not necessarily based on the data of the authors. For example, how would the FLOT1/FLOT2 dimer look if ONLY AlphaFold was used to predict it? Thus, the authors should either truly validate their model – for example by generating a mutation that lacks the predicted “interaction hotspot” and look for interactions or, preferably, by generating a high-resolution model by crystallography – or move it to the supplementary information.

In a next step the authors then use crosslinks from enrichments of early endosomes from FLOT1/2 overexpressing HEK93 cells to validate this hetero-dimer and run native gels, which suggest that some form of higher oligomerization is taking place (4mers, 8mers, 28mers and >36mers). The authors then use this data as an incentive to model a potential FLOT1/FLOT2 heterotrimeric 38-mer. While this model looks admittedly spectacular, it is unfortunately currently not based on data.

It is not possible to obtain exact stoichiometric data from eluates run on a native gel. Also, the cross-links are unable to discriminate between higher oligomers, so for example a 4mer or a 38 mer. Moreover, their own data would suggest a 4mer as the main species – however, the authors are unable to build a tetrameric model.

Their model is therefore not only speculative but also misleading, as it suggests being supported by their own data. It should therefore either be supported by data (for example by obtaining a high-resolution model of the complex by cryoEM or crystallography, or at the very least be further validated, for example by mutating the predicted binding site and show that higher oligomers are no longer formed). Alternatively, it should be removed from the manuscript.

In a last, and again interesting, part of their study, the authors then use these endosomes and MS-based proteomics to identify likely cargo proteins. The authors identify mainly receptors from multiple families and move on to validate three members of the G protein receptor subfamily by Co-IP and Co-immunostaining. As this part of the work is likely the most novel one, it may be helpful to validate additional cargo proteins, at least by Co-IP.

In summary, the authors use MS-based proteomics and biochemical enrichments to comprehensively characterize the protein content of lysosomes and early endosomes and to identify potential new interactors of the V-ATPase and flotillins together with likely cargo proteins. Despite some overstatements, this is an impressive amount of work and in principle suitable for publication in Nature Communications.

Minor Points

1) Cross-links at both FDR 5% and 1% are shown in the supplementary datasets.

The reason for this is unclear to the reviewer - why was not the preferable dataset at 1% FDR cut-off used throughout the manuscript?

In any case and to avoid confusion, it should be clearly indicated throughout the manuscript (in the text or at the respective figure legends) which dataset was used.

2) For the overall cross-linking dataset, it should be indicated how many cross-links were identified consistently, i.e. over multiple independent replicates.

Also, what does "replicate" imply in the context of this manuscript – i.e. biological, fully independent sample or same lysosome enrichment ? Please clarify.

3) Supplementary Table S2: what does "at least three independent runs" imply? - biological replicates including independent cell culture & enrichments or simply multiple MS runs? Real biological replicates are necessary to obtain valid and reproducible data.

4) Fig6 B – "Data dependent acquisition" (DIA) should be data independent acquisition.

Response to reviewer comments

We would like to thank all reviewers for the positive assessment of our work and the very constructive and fair criticism. We addressed all points, the response to individual requests can be found below. We believe that the additional experiments and modifications to the manuscript significantly improved the quality of the manuscript. We hope we were able to address all issues to the full satisfaction of all reviewers.

Reviewer #1 (Remarks to the Author):

The authors have used crosslinking and mass spectrometry on endosomes and lysosomes isolated from human cells to identify protein complexes of these organelles. The follow up with characterization of several of the interactions that they identified as a validation of these results. This includes a PPT1 tetrameric complex; FZD9 interactions with a vATPase subunit; and flotillin interaction with GNB4. They furthermore present data in support of heterodimers of flotillins that may assemble into large multimers to support cargo trafficking and propose latrophilins as a putative cargo of flotillin-dependent trafficking. The experiments are generally performed with a high degree of rigor and interpretation of the results is reasonable. This data should be of a useful resource for investigators of lysosome cell biology. The new insights regarding the proteins mentioned above are intriguing but have not been developed with sufficient depth to confidently demonstrate biological relevance.

However, I recognize that fully defining such relevance is beyond the scope of this manuscript. As a result, I see greatest value as a resource for investigators in the lysosome cell biology community but am less confident about the extent to which each individual finding will stand on its own. I therefore have concerns that the title promises more than the article delivers. I also have some concerns about specific issues that I have summarized below.

R: We would like to thank this reviewer for the positive assessment of our work and the confirmation that our dataset will provide a valuable resource for the community. In order to tone down the title and to adapt it to the actual level of follow up studies performed, we changed it to "Cross-linking of the Endolysosomal System Reveals Potential Flotillin Structures and Cargo".

1. Figure 5 A and B are meant to show partial localization of flotillins to lysosomes. However, the magnification of the images is not adequate to appreciate this message and it is not clear whether this represents flotillin localization to subdomains on lysosomes or just coincidental (and possibly non-specific) overlap.

It is furthermore not clear if the LAMP2 signal is specific as it does not clearly identify individual lysosomes but is found throughout the cytoplasm and is unexpectedly enriched in the nucleus.

R: We would like to thank this reviewer to bring this to our attention. After careful re-evaluation of our images, we realized that the alleged enrichment of LAMP2 in the nucleus was due to a problem with image processing of individual z-stacks due to prisms which were installed in our microscope during a recent upgrade. The rather unspecific patterns of LAMP2 were due to this issue, which, after deconvolution of z-stacks, resulted in an unrealistic increase of size. We apologize for not realizing this earlier. We re-prepared the majority of microscopy samples and re-acquired all microscopy images with a confocal microscope at the core facility microscopy (see methods section), replacing all pictures in

the manuscript. These pictures now show a more realistic pattern of LAMP2 staining. We also assessed the co-localization of LAMP2 and flotillins by image analysis (see Figure 5A, Mander's coefficients and profile plots).

2. Colocalization of flotillin and latrophilin in Figure 7B is also not well developed. It is not clear how much of each signal is specific; how much overlap is coincidental and whether this is occurring on endosomes/lysosomes.

R: Also these images were re-analyzed by confocal microscopy in order to provide higher resolution pictures and to increase the stringency of analysis. We further calculated the Mander's coefficient for the whole cell area and provided profile plots for two different areas of each cell shown. We performed similar analyses for EEA1 to assess the level of unspecific-colocalization in our images.

3. Kaushik references not defined on page 12.

R: The reference has been defined.

Reviewer #2 (Remarks to the Author):

The study of Jasjot Singh and collaborators entitled «Cross-linking of the Endolysosomal System Reveals Flotillin Structures and Putative Cargo» aims to identify the proteins of the lysosomes in HEK293T cells and their interactions. They use the technique of cross-linking mass spectrometry on lysosome enriched fractions, kept intact or permeabilized to favor the penetration of the cross-linking agent. From 2467 unique residue-to-residue cross-links, they identified 270 intra- and 254 inter-protein interactions. The highest numbers of PPI were identified for the V-ATPase, the flotillins, mTORC1 and the syntaxins.

Some global and more specific data of protein-protein interactions from these lysosome enriched fractions (for the V-ATPase and the flotillins) are presented in figure 2. Then they combined the cross-link data with available data from crystallography or generated using AlphaFold, a new tool recently available to propose the structure and assembly of a panel of proteins (alpha-galactosidase A, V-ATPase subunits, and PTP1 dimer and tetramer). Then the authors focused on flotillins, for which they identified the highest number of cross-links. Using a combination of several bioinformatic tools that predict secondary structure, 3D structures combined with the cross-link data, and the integration of known structural features, they propose a model for the flotillin 1/flotillin 2 homodimer.

Then they also use the technique of cross-linking mass spectrometry on early endosomes enriched fractions from cell that were co-transfected with Flag-tagged FLOT1 and FLOT2. They identified 1081 cross-links from 414 unique proteins. The analysis of the FLOT1-FLOT2 cross-links revealed that they are almost similar (80%) to those from lysosome-enriched fractions. Then the authors proposed a model for the oligomerization of flotillins. Finally, they performed proteomic analysis using the early endosome enriched fraction by SPIONs and using immunoprecipitation against Flag-tagged flotillins in order to propose an identification of the flotillin cargos in Hek293 cells.

The data obtained are potentially interesting but raised several questions, in particular why so many proteins identified are non-lysosomal proteins? Why abundant proteins of the lysosome were not found engaged in PPI? Moreover, a number of specific points need to be addressed (see below).

R: We would like to thank this reviewer for the thorough review and positive assessment of our work. Due to the high sensitivity of the mass spectrometer used in this study, large numbers of proteins are identified even though they are present only in very low amounts in the sample. While SPIONs enrichment results in enrichment factors of up to >100-fold

for distinct lysosomal proteins, it does not allow to completely deplete unspecifically enriched proteins, as organelles and proteins interact with the enrichment columns^{1, 2}. Therefore, it is a common phenomenon in the analysis of protein- or organelle-enriched fractions that high numbers of unspecific proteins are identified. This is especially true for lysosomes, as they are, in relation to other organelles, of rather low abundance (lysosomal proteins are estimated to contribute ~ 0.2 % of the cellular protein mass in HeLa cells³) making it necessary to use large amounts of input material for enrichment. It is furthermore conceivable that many of these proteins originate from organelles which are interacting with lysosomes or degraded in them. With respect to the latter, we identified higher numbers of cross-links for non-lysosomal proteins in the disrupted fraction of lysosomes. As disruption of lysosomes enables access of DSSO to lysosomal substrates present in its lumen, this finding is indicative that these proteins may present potential substrates. The protein numbers which we obtained in the SPIONs enrichment experiments described in this manuscript are in good agreement with results from lysosome-enriched fractions from our group and others^{1, 2, 4-6}, indicating a good performance of the experiment (P4L131). Therefore, when a mass spectrometer of similar performance is used, identification of lower protein numbers would rather indicate a lack of instrument performance than an improved enrichment of lysosomes.

Concerning the lack of cross links for high abundant proteins, we would like to thank this reviewer for bringing to our attention that we did not discuss this phenomenon in the text to a sufficient extend. It is a common observation in XL-MS studies, that even high abundant proteins can yield few or no cross links. This is related to several factors: DSSO, which was used for cross linking in this study, reacts mainly with primary amines present in lysine residues and protein N-termini within the distance constraints determined by the length of the DSSO spacer. Furthermore, posttranslational modifications interfere both with cross-linking and subsequent identification of peptides by database searching. Finally, membrane embedded protein domains are almost inaccessible to the cross-linker. Therefore, in order to yield a cross link, two lysine residues of a given protein have to be in a suitable distance and this part of the protein should be neither modified nor membrane embedded. For example, the highly abundant lysosomal membrane proteins LAMP1 and LAMP2 exhibit only few amino acids at their C-terminus which face the cytosol while the luminal part of the proteins is highly glycosylated. Therefore, despite their high abundance (Table S2), no cross links could be identified. In order to bring this to the reader's attention, we added the following statement to the discussion section of the manuscript (P17L518): "The lack of cross-links for highly abundant proteins, such as LAMP1 and LAMP2, could be due to several reasons. For lysosomal luminal proteins, their location largely prevents cross-linking when intact lysosomes are used, due to the lack of membrane permeability of reactive DSSO. Furthermore, both LAMP1 and LAMP2 only contain a short unmodified cytosolic domain (<10 amino acids) while the lysosomal luminal region is highly glycosylated, which interferes with the XL-LC-MS/MS analysis. This is due to a possible interference of glycosylation with the cross-linking reaction, altered fragmentation patterns of glycosylated peptides, and limitations with respect to the incorporation of certain posttranslational modifications during cross-link identification by database searching."

The organization of the article is a bit confusing and the presentation of the different examples is a bit of a catalog. The authors should think about to reorganize the data in figure 2, 3, 4

R: We organized the manuscript according to the different types of information which can be obtained from our cross-linking dataset, as well as the different types of follow-up studies. While it is certainly conceivable to rather organize the information found in figures 2, 3, and 4 based on the individual proteins, it would make it more difficult to explain the reader why the individual experiments were performed. In XL-MS studies, two different types of information can be obtained from a dataset: protein-protein interactions (PPIs) and structural information. Figure 2 (and Figure S2) is only related to PPIs, related analyses,

and follow-up experiments. As we only show a small part of the dataset, this figure is mainly intended to showcase the amount of data present in our dataset and to prove the validity of the identified interactions through follow-up studies. Figure 3 and 4 (as well as Figure S3 and S4), on the other hand, focus on structural information and detailed analyses of vATPase, PPT1 and FLOT1/2. If we would combine both structural and PPI data for vATPase and FLOT in the same figure, it would make it difficult to describe the different aspects of our dataset. For example, the validation of distinct vATPase and FLOT2 PPIs (Figure 2) would be discussed separately in Figures 3 and 4, and therefore in different sections of the text. One of the main purposes of performing these experiments was the validation of PPIs identified in our dataset. Disconnecting these data from the PPI dataset would make it more difficult for the reader to make this connection. We hope that this reviewer can agree with our line of argumentation.

Remarks

1) This study is performed in Hek293T cells. Why having chosen these cells? Will the proteins identified in the lysosomal fraction vary depending on the cells chosen? Would the proteins that have been studied more precisely (flotillins) also be present in other cell types?

R: We chose HEK293 cells, as they allow for the efficient enrichment of lysosomes with a high yield and an excellent amount of intact organelles, which was a prerequisite for this study^{1, 2}. Other cell lines for which we previously performed lysosome enrichment, such as SH-SY5Y or HuH-7 do not allow for such high yields and intact ratios, while the proteome of e.g. HeLa cells seems to be more adapted to the cell line's individual properties². While certainly cell-type specific differences in the abundance of known lysosomal proteins exist, the vast majority of them is present in all human cell lines studied by our group so far. This is not the case for all proteins observed in lysosome-enriched fractions, especially unspecifically enriched ones².

In a previous study, we found both flotillins in lysosome-enriched fractions of 6 different cell lines (HEK293, HeLa, HuH-7, NIH3T3, MEF, and SH-SY5Y)². Furthermore, it is well established that flotillins are highly conserved across organisms and present in nearly every type of vertebrate cell⁷.

2) Preparation method. As the SPIONs technique is the basis for characterization studies of lysosome or early endosome proteins, it is important that all elements to understand the experimental procedures are given. A better description of the method should be given. What is the size, nature of the beads?

The bibliographic reference that is mentioned is not available.

Line 663: DexoMAG40 is unavailable. Is it the same as SphereMAG? Please be sure to provide the details.

R: DexoMAG40 and SphereMAG are both products from Liquids Research Limited and are both described on the same webpage of the manufacturer (https://liquidsresearch.com/en-GB/for_biomedical_applications-57.aspx). In order to see the properties of DexoMAG, the flyer has to be downloaded while SphereMAG is presented on the webpage. We further included the characteristics of the nanoparticles in the methods section (P20L642): "...superparamagnetic iron oxide nanoparticles (SPIONs) solution (DexoMAG40 from Liquid Research Limited, core size: 8 nm, hydrodynamic size: 50 nm, Fe content: 10 mg/ml, coating: dextran 40 kDa)..." A very detailed and thorough description on the SPIONs methodology was published by Walker and Evans⁸, we included the reference in the main text (ref 32, P4L123).

Line 252: the authors said that the length of the DSSA link is around 35Å. In the technical sheet (<https://www.thermofisher.com/order/catalog/product/A33545>) it is mentioned 10,5Å. Why this difference?

R: The distance provided in the product information sheet of DSSO is only related to the spacer arm, meaning the 10.3 Å do not include the NHS groups (the reactive groups at both ends of the spacer). The distance used for inferring of spatial information from cross-linking data is based on the full molecule (spacer and reactive groups) as well as the length of the side chain of the targeted amino acids (in the case of DSSO predominantly lysine residues). It is furthermore important to mention at this point, that the 35 Å present the upper distance limit which can typically still be cross-linked⁹, while also residues in closer proximity will react successfully. Based on experimental data from several studies, e.g.^{10, 11} (both refs are also provided in main text), it is well established that DSSO provides for lysine residues an effective cross-linking range of up to 35 Å, which we also used for this study.

It will be important to show the accumulation of the beads in lysosomes by electron microscopy? or using fluorescent microscopy technique?

R: It is well-established by us and others that SPIONs facilitate an efficient enrichment of lysosomes^{1, 2, 6, 8, 12, 13}. In the course of some of these studies, it was also shown by fluorescence microscopy^{8, 12, 14} and electron microscopy^{13,14-16} that SPIONs localize to lysosomes. We included one citation each for fluorescence and electron microscopy in the text to bring these studies to the reader's attention (P4L123). Furthermore, we show in this study by western blots, enzyme assays, and mass spectrometry, that lysosomal marker proteins are enriched and markers for other organelles are depleted. We therefore hope that this information is sufficient for this reviewer and that it is not necessary to provide additional microscopic investigation for the lysosomal localization of SPIONs.

3) Table S3 and Figure 2:

The authors claim that they generated a protein-protein interaction network containing 70% potential novel protein-protein interactions. But what is puzzling is that the analysis of the inter-links led to the identification of many proteins that are not lysosomal proteins (table S3). Are these proteins false-positive? Or are these proteins there because they undergo a degradation process? Or are they lysosomal protein not identified yet? The fact that many proteins identified by the PPI are non-lysosomal protein has to be discussed in more details.

R: All possibilities provided by this reviewer may be true. In case of interactions between two non-lysosomal proteins, they may either be unspecifically enriched, degraded by lysosomes, or part of a lysosome-associated complex. As soon as a known lysosomal protein is involved in the interaction, it is highly likely that the interaction partner is a so-far unknown lysosomal protein. The reason for this observation can be attributed to the generally low abundance of lysosomal proteins in comparison to other organelles such as the nucleus or mitochondria. This results in a certain abundance of unspecifically enriched proteins relative to specifically enriched ones in lysosome-enriched samples. Based on the data at hand, it is not possible to eliminate unspecifically enriched proteins without risking to remove a potential novel lysosome-located protein which is why we did not remove any of the identified PPIs. We brought this to the reader's attention by adding the following sentence to the results section of the manuscript (P4L140): "Out of the 2,467 unique residue-to-residue cross-links, 524 identifications (270 intra links between different residues of the same protein and 254 inter-links between two different proteins) originated from 111 proteins assigned to the lysosomal compartment (Figure 1E, Figure S1D-H, Table S3), while the remaining cross-links were identified for proteins which are currently not connected to lysosomes, presenting potentially novel interaction partners." As well as in the discussion section (P17L505): "We also identified many cross-links from proteins that are seemingly unconnected to lysosomes. These proteins present possible substrates which were degraded in the lysosomal lumen during the time-point of enrichment, members of protein complexes associated with the lysosomal membrane, or unspecifically enriched

proteins binding to the magnetic columns, which could be due to the large amount of cells utilized as input for lysosome enrichment (64 plates per replicate).”

4) Equally confusing is the fact that abundant lysosome protein families are not identified, such as LAMPs or tetraspanins. This is just mentioned in the discussion but no explanation is provided. How do you explain this result?

R: As explained above, it is not possible to infer the likelihood of observation of cross-links for a certain protein based on its abundance. The decisive factors are its structure, availability of lysine residues, PTMs, and the embedding of protein domains into membranes. We already explained the case of the LAMPs above and added a section to the discussion of the manuscript. For the tetraspanins, a large section of these protein is present in membranes (4 membrane spanning helices), reducing the chance to yield cross-links. If suitable lysine residues would be available in the proteins' cytosolic domains is hard to estimate, as also for most tetraspanins only predicted AlphaFold structures are available and it is not clear from these structures which domains are cytosolic and therefore cross-linker accessible.

Considering this point the title (line 200) « Characterization of the Human Lysosomal Interactome » is a bit of a stretch.

R: We agree with the reviewer and changed the header to “Investigation of Lysosomal Protein-Protein Interactions”.

Also, in the discussion, the authors mention a study using BirA to identify LAMP partners (Go et al, Nature 2021). Although few partners were identified in this overall study, it is more than zero, as was the case in the submitted manuscript.

R: To be honest we do not understand the motivation behind this comment. We did not comment on individual protein/protein interactions or the performance of individual studies, neither did we intend a direct comparison, as indicated by this reviewer. The mention of this study (Go et al) was not intended to highlight individual interaction partners of a specific LAMP, but to make clear that proximity biotinylation approaches in general are not able to provide direct evidence for the nature of interaction between two proteins. Also, in the original study by Go et al. the authors did not discuss individual interactomes from each LAMP, but rather utilized these data together with interaction partners of several other BirA* tagged lysosomal membrane proteins allow for the identification of novel lysosomal or lysosome-interacting proteins. The key point we wanted to make is that their data only indicate that a certain protein resides in the vicinity of the bait protein fused to BirA*. To exemplify this point in the manuscript, we mention the fact that 80 % of interaction partners for LAMP1/2/3 are found for at least two of the three proteins, making it difficult to identify which of the two proteins is the direct interaction partner and which is coincidentally in its proximity (assuming that they do not share 80 % of their interaction partners). Furthermore, even for the unique binding partners, the domains of the proteins interacting with each other cannot be defined. We only used LAMP1/2/3 as an example here, not due to its relation to our dataset. To make this clearer for the reader, we added modified the discussion section accordingly, it reads now as follows (P17L531): “In contrast to cross-linking, proximity biotinylation only allows to determine the presence of a protein within a defined radius relative to the respective fusion construct. It cannot identify, however, if two proteins are directly interacting, or which residues/domains are involved. This is exemplified by a recent study utilizing BirA* fused to eight different proteins including LAMP1, LAMP2, and LAMP3 for investigation of the lysosomal proteome⁷⁰. Comparison of the interaction partners, which were identified for the individual LAMPs, revealed that only 8-18 % were unique, while the others were identified for at least two of them. From these data, it can be inferred that the latter proteins are in close proximity to the lysosomal

membrane, but it cannot be identified with which of the LAMP proteins they directly interact. This is not only the case for these three examples, but for all lysosomal proteins which may be utilized as bait. If a specific cross-link is found for a given lysosomal protein, on the other hand, the respective amino acids have to be within a distance of 35 Å, implying direct interaction. Therefore, compared to proximity biotinylation approaches, the interactome presented in this study provides a level of detail that is unprecedented for the analysis of lysosomal PPIs. .”

Lines 529-532, what do the authors mean? Not clear. What will be more interesting is to compare the advantages and disadvantages of each technique.

R: We believe that we already explained most of this question in response to the previous point of this reviewer. As it seems that our wording is not clear and leads to misunderstandings, we rephrased the text to be clearer and included pros and cons of both approaches (P18L544): “Furthermore, cross-linking allows for the unbiased analysis of PPIs while proximity biotinylation requires expression of fusion proteins, possibly inducing artefacts. The latter approach requires, however, less input material and allows for identification of PPIs irrespective of the presence of lysine residues in a suitable distance, facilitating the identification of a larger spectrum of interaction partners.”

5) Line 207 the authors say that lysosomal PPI « revealed an overrepresentation of nuclear and cytoplasmic/cytoskeletal proteins ». Are these the same proteins identified in the study by Go et al, Nature 2021 considering the lysosomal proteins associated partners?

R: The study by Go et al. is based on distinct bait proteins and their proximity biotinylation interactome. It is therefore difficult to directly compare their dataset to ours. In this context, it is important to take into consideration, that we enriched intact organelles, which will inevitably also result in co-purification of interacting cellular structures. Go and colleagues, on the other hand, investigated individual protein interactomes and used a sophisticated bioinformatic approach to assign which of the identified interaction partners across all 192 baits used in their study are likely to localize to a distinct compartment. As this also included proteins in the cytosol and the nucleus, it is very unlikely that these proteins would be assigned to lysosomes as, even if they interact with the lysosome, the bulk of the protein will still be present at the other location and therefore not be assigned to lysosomes.

Setting this aside, we could only compare interactions found in our dataset for the bait proteins used by Gao et al (8 different lysosomal bait proteins). For these proteins, we were able to identify cross-links with other proteins (inter-links) for two of them, LAMTOR1 and STX7. The total number of cross-links for these proteins with non-lysosomal proteins was below ten. Therefore, a meaningful comparison to the study by Go et al. was unfortunately not possible.

6) Figure 2H and I

Except one case of PPI between Flot2 with another protein (around aa 250), in all cases the PPIs between FLOT1-FLOT2 are located at the same aa sequence than the FLOT1/2-other. How the same region could be involved in FLOT1/FLOT2 interaction and FLOT1/2-other proteins interactions?

R: The presence of a cross-link indicates that a certain fraction of the respective proteins interacts, but neither that this interaction is exclusive, nor that only monomers of the individual proteins interact with each other. With respect to the examples at hand, it is certainly conceivable, that the identified interaction partners are binding either to FLOT1/FLOT2 heterodimers, homodimers, or monomers, which all have been demonstrated to exist *in vivo*¹⁷ and for which several examples are available in the literature. It could also well be, that one pool of FLOT1 interacts with FLOT2 while another pool interacts with the other identified proteins. Furthermore, interaction of the alpha helical

regions of FLOT1 and FLOT2 does not imply that the respective region is completely blocked for interactions with other proteins. When considering these facts, the data are highly indicative that the regions highlighted in Figure 2H/I possess properties which facilitate interaction with other proteins, but should not be contradictory.

Lanes 227-229 : not clear, to better explain.

R: We changed to wording of the manuscript to be clearer, it reads now as follows (P6L213):
“The fact, that we found FLOT1/FLOT2 inter-links across the whole region of the proteins shows that no sequence-dependent bias towards the cross-link reaction or detectability exists. Therefore, the of the majority of PPIs with protein other than FLOT1/FLOT2 to a distinct section of the proteins suggests the presence of FLOT1/FLOT2 interaction hotspots.”

7) Why the authors splited the data on the V-ATPase at two distinct parts of the results (lines 210-218 and 259-272)?

Idem concerning the flotillin data. Why having split them between figure 2 and 4?

R: We already commented on this remark in our answers to the general comments above. The key point is that we discuss our dataset based on its individual contents and therefore clearly separate the identification/characterization of protein-protein interactions and structures. Combining all vATPase or FLOT data in one figure each would not have allowed to keep this structure of the article, and we would have to jump back and forth between individual aspects. We believe that our current choice of organization is easier to follow for the reader and hope that this reviewer can agree with us in this point.

8) Line 325 and line 584: the authors mentioned a Flot2 N-term NMR structure. No reference is provided. Reference #54 does not present such data.

R: The N-terminal NMR structure of Flot2 was only deposited in the protein databank (PDB) with the identifier 1WIN (<https://www.rcsb.org/structure/1WIN>). There is no associated manuscript to date. Reference 56 relates to the difficulty of purification of flotillins and not the structure. In order to make this clearer for the reader, we adjusted the text, it reads now as follows (P10L301): “It is well-established, that FLOT1 and FLOT2 form heterodimers with a 1:1 stoichiometry 55, but only partial structural information is available from NMR analyses of the N-terminal region of mouse Flot2 (PDB identifier 1WIN). Other experimental data on FLOT structures are not available, as purification of the full-length proteins is problematic 56.”

9) Lines 340-342: The authors selected model four among the 5 models for the FLOT1-FLOT2 heterodimer obtained using ColabFold. This is not clear why? Couldn't these different models co-exist? The added value of the analysis performed using cross-linking and proteomics is not emphasized.

R: We apologize for being not clearer with respect to this issue. The top five ColabFold models of the FLOT1-FLOT2 complex are shown in the supplementary information in Figure S4. All generated models satisfy the identified cross-linking restraints and could therefore, in theory give rise to the identified cross-links. Only model 4, however, had no unstructured regions and adequate helical-helical interaction between FLOT1 and FLOT2. This led to our choice to show model 4 in main Figure 4 and to use it for highlighting the interactions and features for illustrative purposes. Specifically, we do not claim that this model is more likely to be accurate than the other four models given the lack of further evidence. To address this point, we changed the caption of Figure 4 to "Heterodimeric model of FLOT1-FLOT2 interaction generated by ColabFold" and added the following statement to the figure legend of Figure S4: “While all models fulfil the distance constraints of the identified cross-

links, model 4 was the only one without unstructured regions and full alignment of the α -helical structures. Therefore, model 4 was selected as most likely structure.”

These software alone are already a revolution for obtaining 3D structures of proteins and their complexes. What is the difference between the FLOT1-FLOT2 heterodimer obtained by ColbFold alone compared to the one obtained by combining ColabFold and the cross-link data?

R: We are sorry that this did not become clear in the manuscript. We have expanded the corresponding section in the manuscript as follows and hope this provides sufficient explanation of the differences (P10L319): “Subsequently, we built heterodimeric models of FLOT1, FLOT2, and their interaction using ColabFold¹⁸, a variant of AlphaFold. For additional post-modeling validation with experimental data, we mapped the crosslinking restraints on the five resulting models. All generated models satisfied all experimentally identified restraints based on the cross-links, increasing our confidence in these models.”

10) Could it be possible to analyze the impact of the mutation of the tyrosine phosphorylation sites, which were shown to be crucial for flotillin-mediated endocytosis and FLOT1/FLOT2 interaction (Y160 in FLOT1 and Y163 in FLOT2), on the FLOT1-FLOT2 heterodimer obtained by ColbFold?

R: This is indeed an interesting question. AlphaFold has not been trained or validated for predicting the effect of mutations. In particular, it is not expected to capture the effect of point mutations that destabilize a protein. A recent study, which is currently available on BioRxiv¹⁹, attempted to predict the impact of single mutations on protein stability and function using AlphaFold. As a readout, the authors tracked the residue confidence score of predicted local distance difference test (termed pLDDT) on a large-scale dataset of reported mutations. They reported a very weak or no correlation between AlphaFold output metrics and the change of protein stability or fluorescence¹⁹. Nevertheless, we used AlphaFold for modeling FLOT1-FLOT2 interaction in case of the suggested mutations (Y160 in FLOT1 and Y163 in FLOT2), and – predictably - the model looks basically identical as the wildtype model. We therefore did not include these data in the manuscript.

11) Immunostaining for flotillins.

Flotillins are well known to be located in lysosomes/late endosomes. The images presented in figures 5 and 7, S6 and S7 are wired. In Hela cells these staining do not reveal vesicular structures such as lysosomes/late endosomes. The staining's reveal filamentous structures but not vesicles. Being able to see colocalizations with these types of images is not the guaranty of the real co-localization in structures such as LE.

R: We would like to thank this reviewer for bringing this mistake to our attention and we would like to apologize for not realizing it ourselves in the first place. We performed several tests with our microscope and it turned out that we had a technical issue which caused the stainings to appear filamentous rather than punctate. This was related due to issues with a prism included in a recent upgrade of our microscope resulting in image distortion for individual z-stacks. In order to address this problem, we re-imaged all samples using a confocal microscope at our microscopy core facility. The respective new figures are now included in the main figures 5 and 7 as well as supplementary figures 5-9.

It will be important to try other fixation/permeabilization methods and to really identify flotillins upon expression of fluorescently-tagged versions.

R: We tried several other fixation/permeabilization methods and the resulting images are included in Figure2_Review_Only. These analyses revealed that fixation with ice cold methanol was superior, which is also in accordance to published data which indicate that

methanol fixation is ideal for accessibility of flotillin antigens for subsequent antibody staining²⁰. As mentioned already, in the course of these experiments, we identified the microscope as source of error for the filamentous appearance. Utilization of fixation with ice cold methanol in combination with the confocal microscope of the core facility microscopy yielded satisfying results. We further confirmed correct localization by transfection of cells with FLOT1-GFP and/or FLOT2-mCherry followed by co-staining with LAMP2 (P12,L377). Co-localization analysis (Mander's coefficient and line plots) of FLOT1-GFP/FLOT2-mCherry, FLOT1-GFP/LAMP2 and FLOT2-mCherry/LAMP2 showed overlapping populations of the individual proteins, as expected.

Also comparing the images of Figure 5A to those of Figure S5A for HeLa cells in particular, the distribution of the FLOT/FLAG (Fig S5A) does not match with the immunostaining (Fig 5A). The quantification of the colocalization is required.

R: We calculated the Mander's coefficient for all images and included it together with the images. Furthermore, we generated line plots to investigate the co-localization of individual vesicles. As consequence of the markedly increased picture quality due to analysis with a confocal microscope the co-localization of individual vesicles can be observed much better.

The inset are very small and have to be enlarged. Showing so many cells is not important, I will be helpful to show only 2/3 enlarged cells and to associate a colocalization analysis.

R: We increased the size of all insets shown in the manuscript.

12) In the reference mentioned line 581 (Kauchik), it was shown electron microscopy images showing for example the co-localization of LAMP2 and flotillin1. This makes again puzzling the absence of LAMP2/FLOT1 cross-links.

R: As already discussed above, the LAMPs possess only a rather short cytosolic domain. In the case of LAMP2, these are the last 10-12 amino acids at its C-terminus²¹ which only contain a single lysine residue in close proximity to the transmembrane domain. Therefore, it is highly unlikely that a lysine residue of FLOT1 will be in the right distance to enable a cross-link between these two proteins. Furthermore, we would like to mention at this point that co-localization of FLOT1 and LAMP2 is no direct proof that they interact with each other, it is possible that another protein is required to form this interaction.

The authors said (lines 581-583) « Taken together, these data provide strong evidence that the lysosomal localization of GNB4 is mediated through interaction with the flotillins. » At the same time they said that a yeast two-hybrid screen revealed a direct interaction of GNB4 with LAMP2. Why flotillins should be involved then?

R: Also, in this case the situation may not be as clear cut as implied in this comment which suggests that only a single interaction of GNB4 takes place in this context. It is certainly conceivable that GNB4 interacts not mutually exclusive with either LAMP2 or FLOT1/2 but that it could interact with both of them, e.g. through different domains. The yeast two hybrid screen indicates interaction of LAMP2 and GNB4, it does not, however, exclude further interaction of GNB4 with FLOT1/2 or that even a trimeric complex of all proteins is formed in mammalian cells. While one GNB4 molecule interacts with LAMP2, another one could also interact with FLOT1/2. It could also be possible that one of these proteins recruits GNB4 to lysosomes where it will then interact with the other one. Therefore, in our eyes the interaction of LAMP2 and GNB4 does not exclude further interaction at all, and, as stated in the manuscript, rather confirms the validity of our data as it provides further proof that GNB4 is indeed located at lysosomes. In order to make it more clear that the localization of GNB4 at lysosomes and the involvement of FLOT presents a situation which is less clear than our initial statement suggested, we changed the conclusion at the end of

this paragraph as follows (P19L600): “Taken together, these data provide further evidence that GNB4 is located at lysosomes and further imply that flotillin could be functionally connected to it.”

13) Why having analyzed proteins in EE?

R: FLOT1/2 are best characterized for their role in endocytosis and known to be located at the plasma membrane and early endosomes, which was also the case in our microscopy analyses. This raised the question, if there are structural differences between the endosome and lysosome interacting populations of these proteins, which would imply an additional regulatory layer. The underlying logic of our analyses was to investigate whether FLOT1/2 localizing on early endosomes form the same structure as on lysosomes, see P12L372: “In order to elucidate if the structural assembly of FLOT1 and FLOT2 differs between early endosomes and lysosomes, we performed cross-linking of early endosome-enriched fractions.”

14) How the model in figure 5I was obtained? No indication neither in the text nor the figure legend are mentioned.

R: Based on the comments from this reviewer and reviewer 3, we decided to remove our predicted structure of the 38mer flotillin from the manuscript together with all related information.

15) How the author try to model the tetrameric FLOT1-FLOT2 complex? They need to explain that and to discuss on which basis this model was proposed.

R: We tried to model the tetrameric FLOT1-FLOT2 complex using two different approaches: restraint-based docking and AlphaFold-multimer. For restraint-based docking, we used the identified cross-links to build the interaction. However, there were two challenges to this approach. A) we did not know the correct topology or the arrangement of the subunits within the tetramer, as the cross-links identified do not allow to identify if they originate from a dimeric or higher mono/heteromeric structure. B) Because of the rigid-body docking approach, the docking software brought the subunits close to each other based on the identified cross-links, but it didn't achieve a well-formed interface between the PhB domains that are known to be interacting, as they were not covered by any cross-links. For AlphaFold-multimer modeling, the resulting models have very low confidence scores. Furthermore, the most likely model features a major bend in its main helix (see Figure1_Review_Only), which is not in accordance to the lack of long-distance cross-links which are clearly opposing such structure (see P19L609). To clarify this point we added the following statement in the manuscript (P12L404): “...,we further attempted to model the tetrameric structure using either restraint-based docking or AlphaFold multimer. This did, however, not result in a convincing structural model. .” We do not think that it would be informative for the reader to further elaborate our attempts and would only unnecessarily inflate the manuscript. Therefore, we did not go further into detail in the text.

Line 602 they talk about « an abundant tetrameric assembly », this appears contradictory?

R: This section has been removed from the manuscript.

16) What is the impact of mutation of the Y residues proposed to be involve in FLOT1/FLOT2 interaction on the model figure 5I?

R: As pointed out above, we decided to remove this model from the manuscript.

17) Why the authors said « A potential function for such an assembly could be the formation or transport of flotillin-positive endosomes »? This has to be explained.

R: As pointed out above, we decided to remove this model from the manuscript.

18) Amongst the 300 proteins identified by co-IP with flotillins and proposed as flotillin cargos, how many are found in the lysosome-enriched fraction?

R: We would like to thank this reviewer for bringing this possible comparison of our datasets to our attention. We further would like to emphasize that we did not analyze interaction partners of FLOT1/2 by co-IP but we enriched intact early endosomes which also contained FLOT1/2 on their surface. Out of the proteins identified in these analyses and defined as putative FLOT1/2-endosome cargo, 189 proteins were also identified in the global (no cross-linking) proteomic analysis of lysosome-enriched fractions. We modified the manuscripts as follows to bring this to the reader's attention (P14L450): "The enriched population, which presents potential cargo of FLOT1-/FLOT2-positive early endosomes, consists of 328 proteins (Supplementary Table 7). Of these, 189 were also identified in the lysosome-enriched fractions (Supplementary Table 2), possibly indicating their destination for lysosomal degradation.

19) From table S7 it is not obvious that many proteins are membrane or transmembrane proteins as shown in figure S6F. In this table many proteins are RNA-binding proteins, DNA-binding proteins. Could these proteins be flotillin cargo? Why they are not present in figure 6C?

R: We would like to thank this reviewer for bringing this to our attention. To make it clearer for the reader to which categories our putative cargo molecules belong, we included an additional sheet which contains how many proteins were assigned to a certain category (Table S7, sheet 5). As can be seen in this analysis, DNA/RNA proteins indeed present a prominent fraction with 9 and 30 proteins each. Compared to proteins related to membranes, vesicles, and such related to receptors/extracellular space (in total >150 proteins), however, their numbers are significantly lower. Whether the identified DNA/RNA binding proteins are truly interacting with flotillin is a subject which would require further evaluation. As the main purpose of these experiments was to identify cargo which originates from flotillin mediated endocytosis at the plasma membrane, we believe that DNA/RNA binding proteins seem to be rather unlikely, and could present a contamination, e.g. through unspecific interaction with the affinity beads or association to flotillins on the outside of endosomes. Therefore, we did not further comment on them in the manuscript. Concerning Figure 6C and Figure S7G, we only included GO categories which were statistically significant overrepresented. In this context, we only identified proteins in the category "cellular component" to be overrepresented and the respective members are included in these figures.

20) Concerning the discrimination of « protein populations that were enriched/depleted in FLOT1-/FLOT2-positive endosomes relative to the total cellular pool of early endosomes », the experimental procedure should be better explain. Where the data of the proteins depleted in the FLOT1-/FLOT2-positive endosomes are found.

R: The experimental procedures for these experiments are included in the section statistical evaluation. As we see that this may be difficult to identify, we changed to title to "Identification of putative FLOT-endosome cargo by statistical analysis" (P25L942). All down-regulated proteins are also included in Table S7. As we can see that it may be too laborious for the reader to identify these proteins based on the protein ratios and p-values provided, we included an additional sheet which includes all significantly depleted proteins (sheet 4).

Minor points

- Lines 575 and 581: the references were not included

R: The missing references have been included. -

Line 854: what is PDB: 1WIN?

R: As already explained above, PDB stands for the Protein Data Bank, 1WIN is the mouse FLOT2 NMR structure. As we already explained this in the main text in the revised version of the manuscript (P10L370) we did not add additional information to this section.

Reviewer #3 (Remarks to the Author):

In their study Singh et al. investigate lysosomes and early endosomes by MS-based proteomics. In a major experimental effort, the authors use multiple MS-based proteomic workflows, namely cross-linking mass spectrometry (XL-LC-MS/MS), to identify the protein content and protein-protein interactions (PPIs) from enriched lysosomes and early endosomes. In a first part the authors use superparamagnetic iron oxide nanoparticles (SPIONS) to enrich lysosomes followed by cross-linking of intact and disrupted lysosomes. By comparison with known lysosomal proteins, the authors can convincingly show that they are able to enrich lysosomes and by applying an established cross-linking workflow - which the authors apply the first time to lysosomes - they are able to detect cross-links on multiple proteins, including in particular the V-ATPase and flotillins. Here, the authors identify potentially novel interactors – namely FZD9 for the ATPase and GNB4 for FLOT1 and 2 – which the authors subsequently validate by Co-IP. The authors then move on to use the identified cross-links for structural refinement of the V-ATPase and the PPT1 complex, which the authors convincingly show to likely exist as a tetramer within lysosomes. This part of the work is very solid and clearly increases our knowledge about lysosomes.

R: We would like to thank this reviewer for the positive assessment of our work.

In a next step the authors then use their cross-linking data and a full-length structural monomeric model from AlphaFold to generate a dimeric model of the FLOT1/FLOT2 dimer using HADDOCK (FLOT1/FLOT2 was known previously to form a dimer). The authors then move on to make precise predictions based on this model. However, the model suggests a level of detail that is not necessarily based on the data of the authors. For example, how would the FLOT1/FLOT2 dimer look if ONLY AlphaFold was used to predict it?

R: We thank this reviewer to draw our attention to this lack of clarity in the manuscript. The individual FLOT1 and FLOT2 monomeric models were solely based on predictions of AlphaFold. As AlphaFold does not provide any information on the dimeric structures, we utilized these monomeric models to predict dimers using ColabFold, a variant of AlphaFold²². This yielded several potential outcomes, on which we mapped the distance constraints of the identified cross-links. The top 5 models of this process are displayed in Figure S4. In order to make it clearer for the reader, we added the following statement to the manuscript (P10L318): “Subsequently, we built heterodimeric models of FLOT1, FLOT2, and their interaction using ColabFold¹⁸, a variant of AlphaFold. For additional post modeling validation with experimental data, we mapped the crosslinking restraints on the five resulting models. All generated models satisfied all experimentally identified restraints based on the cross-links, increasing our confidence in these models.”

Thus, the authors should either truly validate their model – for example by generating a mutation that lacks the predicted “interaction hotspot” and look for interactions or, preferably,

by generating a high-resolution model by crystallography – or move it to the supplementary information.

R: We would like to differentiate this point into two aspects.

For the validation of the FLOT1-FLOT2 heterodimeric ColabFold model, we believe that the information provided by our cross-links is sufficient. We identified 29 unique FLOT1/FLOT2 cross-links, containing 22 inter-links among both proteins, all of which were identified in >3 replicates, providing a very strong evidence. These cross-links, which are distributed along the entire C-terminal helix, are clearly indicative of an extended interaction mode (Figure 4C, S4C), a result which we also obtained by integrative modeling without the input from AlphaFold (Figure 4B).

It was shown previously, that the purification and crystallization of full-length FLOT1/FLOT2 is problematic²³. Therefore, we did not attempt to crystallize the proteins but generated mutants for the proposed interaction hotspot, lacking amino acid residues 200-300 (termed $\Delta 100$ in manuscript) for both FLOT1 and FLOT2 (for details see P20L693). Co-transfection of these mutated versions showed a strong reduction for binding of GNB4 compared to the full-length protein, confirming the presence of the interaction hot spot (Figure 4D). Furthermore, the truncated versions of FLOT1/FLOT2 failed to efficiently form higher order oligomeric structures, as determined by SDS-PAGE analysis of cross-linked FLOT1/FLOT2 enriched samples (Figure 5H). Deletion of this interaction hot spot in both FLOT1 and FLOT2 did not only abrogate interaction with GNB4, but also interfered with the formation of higher-order assemblies, providing further evidence for a prominent role in the formation of PPIs for this region of the flotillins (P12L406):” We further investigated in how far the deletion of the putative interaction hot spot influences the assembly of higher order structures by co-transfection of both FLOT1/2 deletion constructs followed by IP, cross-linking and SDS-PAGE. Western blot analyses of these samples indicate that the absence of the interaction hot spot also influences the formation of higher-order assemblies migrating at sizes >180 kDa (Fig. 5h)”.

In a next step the authors then use crosslinks from enrichments of early endosomes from FLOT1/2 overexpressing HEK93 cells to validate this hetero-dimer and run native gels, which suggest that some form of higher oligomerization is taking place (4mers, 8mers, 28mers and >36mers). The authors then use this data as an incentive to model a potential FLOT1/FLOT2 heterotrimeric 38-mer. While this model looks admittedly spectacular, it is unfortunately currently not based on data. It is not possible to obtain exact stoichiometric data from eluates run on a native gel. Also, the cross-links are unable to discriminate between higher oligomers, so for example a 4mer or a 38 mer.

Moreover, their own data would suggest a 4mer as the main species – however, the authors are unable to build a tetrameric model.

Their model is therefore not only speculative but also misleading, as it suggests being supported by their own data. It should therefore either be supported by data (for example by obtaining a high-resolution model of the complex by cryoEM or crystallography, or at the very least be further validated, for example by mutating the predicted binding site and show that higher oligomers are no longer formed). Alternatively, it should be removed from the manuscript.

R: Concerning the 38mer structure, we were not able to obtain a sufficiently pure fraction to perform cryoEM experiments. It was shown previously, that the purification and crystallization of full-length FLOT1/FLOT2 is problematic²³, which is the reason why we did not further attempt to perform such experiments. As our model contains several interaction interfaces between both flotillins in all parts of the proteins, mutation of all sites would result in a highly artificial molecule. It would therefore be questionable in how far the removal of interacting amino acids or additional effects would influence the behavior of the proteins. Therefore, we decided to remove the proposed 38mer structure from the manuscript.

With respect to the tetrameric FLOT1-FLOT2 complex, we tried to model its structure using two different approaches: restraint-based docking and AlphaFold-multimer. For restraint-based docking, we used the identified cross-links to build the interaction. However, there were two challenges to this approach. A) We did not know the correct topology or the arrangement of the subunits within the tetramer. B) Because of the rigid-body docking approach, the docking software brought the subunits close to each other based on the identified cross-links, but it didn't achieve a well-formed interface between the PhB domains that are known to be interacting (but were not covered by any cross-links). For AlphaFold-multimer modeling, the resulting models have very low confidence scores. Furthermore, the most likely model features a major bend in its main helix (see Figure1_Review_Only), which is not in accordance to the lack of long-distance cross-links which are clearly opposing such structure (see P19L609). To clarify this point we added the following statement in the manuscript (P12L404): "Our attempts to model the tetrameric structure using either restraint-based docking or AlphaFold multimer did not result in a convincing structural model."

In a last, and again interesting, part of their study, the authors then use these endosomes and MS-based proteomics to identify likely cargo proteins. The authors identify mainly receptors from multiple families and move on to validate three members of the G protein receptor subfamily by Co-IP and Co-immunostaining. As this part of the work is likely the most novel one, it may be helpful to validate additional cargo proteins, at least by Co-IP.

R: In order to validate additional cargo proteins, we performed co-IPs of FLOT1 and investigated several candidate proteins which were identified in our dataset by western blotting. We were able to show interaction for 8 additional proteins including 6 receptors, increasing the list of IP-validated putative FLOT endosome cargo proteins to 11 (including the Latrophilins). These data are included in Figure 7D and mentioned in the text at P15L471." For a subset of putative FLOT-positive endosome cargo proteins, we performed FLOT1 co-IPs to investigate a potential direct interaction. In total, we were able to IP eleven proteins with FLOT1 (Fig. 7a and c), encompassing eight receptors, two membrane proteins (PLSCR1 and VANGL1), and one protein interacting with a receptor (SMAD3)."

In summary, the authors use MS-based proteomics and biochemical enrichments to comprehensively characterize the protein content of lysosomes and early endosomes and to identify potential new interactors of the V-ATPase and flotillins together with likely cargo proteins. Despite some overstatements, this is an impressive amount of work and in principle suitable for publication in Nature Communications.

R: We would like to thank this reviewer for the positive assessment of our manuscript.

Minor Points

1) Cross-links at both FDR 5% and 1% are shown in the supplementary datasets. The reason for this is unclear to the reviewer - why was not the preferable dataset at 1% FDR cut-off used throughout the manuscript? In any case and to avoid confusion, it should be clearly indicated throughout the manuscript (in the text or at the respective figure legends) which dataset was used.

R: While filtering at 1% FDR results in a more stringent list of identified cross-links, it also significantly reduces the number of identifications. In our dataset, the number of identified cross-links is reduced by ~28% when filtering with 1% instead of 5% FDR (see Table S2). Therefore, it is very likely that we are also losing a significant number of true positives. This is the reason why we decided to include both datasets (see also Table S3). As it is clearly indicated which data were identified at which FDR, the reader will be able to assess the trustworthiness of individual identifications. Additionally, we added the FDR in the main manuscript when describing the XL dataset (P4L140): "Analysis of the XL-LC-MS/MS

dataset with XlinkX38 resulted in the assignment of 6,580 cross-link spectral matches, originating from 4,294 cross-linked peptides at a false discovery rate (FDR) of 5% (Supplementary Table 3). “

2) For the overall cross-linking dataset, it should be indicated how many cross-links were identified consistently, i.e. over multiple independent replicates.

R: Venn-diagrams including this information are now included in the supplementary material in Figure S2F.

Also, what does “replicate” imply in the context of this manuscript – i.e. biological, fully independent sample or same lysosome enrichment? Please clarify.

R: Individual biological replicates relate to fully independent samples, i.e. individual lysosome/endosome enrichment experiments from different batches of cells differing with regard to their passage number. We added this information to the methods section (P20L660).

3) Supplementary Table S2: what does “at least three independent runs” imply? - biological replicates including independent cell culture & enrichments or simply multiple MS runs? Real biological replicates are necessary to obtain valid and reproducible data.

R: In these analyses, three independent biological replicates were run for both disrupted and intact lysosomes totaling 6 experiments. “At least three biological runs” refers to one of these 6 experiments, referring to independent cell culture and enrichment experiments.

4) Fig6 B – “Data dependent acquisition” (DIA) should be data independent acquisition. R: The error has been corrected.

References

1. Singh, J. *et al.* Systematic Comparison of Strategies for the Enrichment of Lysosomes by Data Independent Acquisition. *J. Proteome Res.* **19**, 371-381 (2020).
2. Akter, F., Ponnaiyan, S., Kögler-Mohrbacher, B., Bleibaum, F., Damme, M., Renard, B.Y., Winter, D. Multi cell line analysis of lysosomal proteomes reveals unique features and novel lysosomal proteins. *bioRxiv* (2020).
3. Itzhak, D.N., Tyanova, S., Cox, J. & Borner, G.H. Global, quantitative and dynamic mapping of protein subcellular localization. *eLife* **5**, e16950 (2016).
4. Mosen, P., Sanner, A., Singh, J. & Winter, D. Targeted Quantification of the Lysosomal Proteome in Complex Samples. *Proteomes* **9** (2021).
5. Abu-Remaileh, M., Wyant, G.A., Kim, C., Laqtom, N.N., Abbasi, M., Chan, S.H., Freinkman, E., Sabatini, D.M. Lysosomal metabolomics reveals V-ATPase- and mTOR-dependent regulation of amino acid efflux from lysosomes. *Science* **358**, 807–813 (2017).
6. Ponnaiyan, S., Akter, F., Singh, J. & Winter, D. Comprehensive draft of the mouse embryonic fibroblast lysosomal proteome by mass spectrometry based proteomics. *Sci. Data* **7**, 68 (2020).
7. Rivera-Milla, E., Stuermer, C.A.O. & Málaga-Trillo, E. Ancient origin of reggie (flotillin), reggie-like, and other lipid-raft proteins: Convergent evolution of the SPFH domain. *Cell. Mol. Life Sci.* **63**, 343–357 (2006).
8. Walker, M.W. & Lloyd-Evans, E. A rapid method for the preparation of ultrapure, functional lysosomes using functionalized superparamagnetic iron oxide nanoparticles. *Methods Cell Biol* **126**, 21-43 (2015).
9. O'Reilly, F.J. & Rappsilber, J. Cross-linking mass spectrometry: Methods and applications in structural, molecular and systems biology. *Nat. Struct. Mol. Biol.* **25**, 1000– 1008 (2018).
10. Kastritis, P.L. *et al.* Capturing protein communities by structural proteomics in a thermophilic eukaryote. *Mol. Syst. Biol.* **13**, 936 (2017).
11. Klykov, O., van der Zwaan, C., Heck, A.J.R., Meijer, A.B. & Scheltema, R.A. Missing regions within the molecular architecture of human fibrin clots structurally resolved by XL-MS and integrative structural modeling. *Proc. Natl. Acad. Sci. U.S.A.* **117**, 1976–1987 (2020).
12. Lee, S.H. *et al.* Endocytic trafficking of polymeric clustered superparamagnetic iron oxide nanoparticles in mesenchymal stem cells. *J Control Release* **326**, 408-418 (2020).
13. Tharkeshwar, A.K. *et al.* A novel approach to analyze lysosomal dysfunctions through subcellular proteomics and lipidomics: the case of NPC1 deficiency. *Sci Rep* **7**, 41408 (2017).
14. Le, T.S. *et al.* Quick and Mild Isolation of Intact Lysosomes Using Magnetic-Plasmonic Hybrid Nanoparticles. *Acs Nano* **16**, 885-896 (2022).
15. Xu, H. *et al.* Differential internalization of superparamagnetic iron oxide nanoparticles in different types of cells. *J Nanosci Nanotechnol* **10**, 7406-7410 (2010).
16. Chen, C.C. *et al.* Simple SPION incubation as an efficient intracellular labeling method for tracking neural progenitor cells using MRI. *PLoS One* **8**, e56125 (2013).
17. Solis, G.P. *et al.* Reggie/flotillin proteins are organized into stable tetramers in membrane microdomains. *Biochem. J.* **403**, 313–322 (2007).
18. Mirdita, M., Ovchinnikov, S. & Steinegger, M. ColabFold - Making protein folding accessible to all. *bioRxiv* (2021).
19. Pak, M.A. *et al.* Using AlphaFold to predict the impact of single mutations on protein stability and function. *bioRxiv*, 2021.2009.2019.460937 (2021).

20. Riento, K., Frick, M., Schafer, I. & Nichols, B.J. Endocytosis of flotillin-1 and flotillin-2 is regulated by Fyn kinase. *Journal of cell science* **122**, 912-918 (2009).
21. Rout, A.K., Strub, M.P., Piszczek, G. & Tjandra, N. Structure of transmembrane domain of lysosome-associated membrane protein type 2a (LAMP-2A) reveals key features for substrate specificity in chaperone-mediated autophagy. *The Journal of biological chemistry* **289**, 35111-35123 (2014).
22. Mirdita, M., Ovchinnikov, S. & Steinegger, M. ColabFold - Making protein folding accessible to all. *bioRxiv*, 2021.2008.2015.456425 (2021).
23. Dempwolff, F. *et al.* Super Resolution Fluorescence Microscopy and Tracking of Bacterial Flotillin (Reggie) Paralogs Provide Evidence for Defined-Sized Protein Microdomains within the Bacterial Membrane but Absence of Clusters Containing Detergent-Resistant Proteins. *PLoS Genet.* **12**, e1006116 (2016).

REVIEWERS' COMMENTS

Reviewer #1 (Remarks to the Author):

The authors have addressed my major concerns relating to images/microscopy. The data relating to colocalization is much more compelling.

One seeming discrepancy that the authors should consider addressing is that when discussing LAMP1/LAMP2 they suggest that the lack of cross-links within luminal domains likely reflects that lack of membrane permeability of the cross-linking reagent. However, it is later on shown that this same strategy successfully yielded crosslinks for PPT1 even though it is a luminal protein. This leads me to wonder if other issues are at play for LAMP1/2 or whether it is possible that not all PPT1 is inside lysosomes. This should not require additional experiments. An explanation in the text should be sufficient.

Reviewer #2 (Remarks to the Author):

The changes made are convincing and so are the answers to the questions asked.

I think it is a pity to remove the multimeric structure of flotillines. It reduces the scope of the message of the article.

I request a correction in the manuscript regarding point 18.

Indeed, the end of the sentence P14L450 has to be modified (degradative has to be remove). Targeting a protein to a late endosome/lysosome (LAMP2-positive) is not always for the purpose of degradation. These vesicles include several subpopulations that we do not currently know how to discriminate with the existing tools. Some of them will have degradation functions while others, following fusion with the plasma membrane, will have secretion or protein recycling functions.

Reviewer #3 (Remarks to the Author):

My points have been sufficiently addressed and I support the publication of the manuscript.

Reviewer #1 (Remarks to the Author):

The authors have addressed my major concerns relating to images/microscopy. The data relating to colocalization is much more compelling.

R: We would like to thank this reviewer for the positive assessment of our experiments performed during the revision of the manuscript.

One seeming discrepancy that the authors should consider addressing is that when discussing LAMP1/LAMP2 they suggest that the lack of cross-links within luminal domains likely reflects that lack of membrane permeability of the cross-linking reagent. However, it is later on shown that this same strategy successfully yielded crosslinks for PPT1 even though it is a luminal protein. This leads me to wonder if other issues are at play for LAMP1/2 or whether it is possible that not all PPT1 is inside lysosomes. This should not require additional experiments. An explanation in the text should be sufficient.

R: This is a very good point and we would like to apologize for not being clearer with respect to this topic. In case of disrupted lysosomes, we should certainly see cross-links for all proteins which are located in the lysosomal lumen, also for LAMP1/2. We currently believe that the lack of cross-links is rather related to the high amount of protein glycosylation which is present on the luminal domains of these proteins. We rephrased the discussion section in order to make this clear, it reads now as follows (L411): "This could be due to several reasons: For the cross-linking of intact lysosomes, the lack of membrane permeability of reactive DSSO prevents the analysis of lysosomal luminal proteins or domains (in case of membrane proteins), and LAMP1/2 only contain a short cytosolic domain (~10 amino acids). While analysis of disrupted lysosomes should allow for coverage of luminal proteins/domains, as was e.g. the case for PPT1, glycosylation of amino acids could be the reason for the lack of identified cross-links for certain proteins, as the lysosomal luminal regions of LAMP1/2, as well as CTSD, are highly glycosylated. Underlying reasons could be interference of glycosylation with the cross-linking reaction, proteolytic digestion, or its influence on peptide fragmentation. Furthermore, in order to allow for the identification of modified cross-linked peptides by database searching, the molecular weight of each modification needs to be defined. This further prevents the identification of glycosylated peptides if no special emphasis is put on their analysis."

Reviewer #2 (Remarks to the Author):

The changes made are convincing and so are the answers to the questions asked. I think it is a pity to remove the multimeric structure of flotillins. It reduces the scope of the message of the article.

R: We would like to thank this reviewer for the positive assessment of our modifications to the manuscript.

I request a correction in the manuscript regarding point 18. Indeed, the end of the sentence P14L450 has to be modified (degradative has to be remove). Targeting a protein to a late endosome/lysosome (LAMP2-positive) is not always for the purpose of degradation. These vesicles include several subpopulations that we do not currently know how to discriminate with the existing tools. Some of them will have degradation functions while others, following fusion with the plasma membrane, will have secretion or protein recycling functions.

R: This is a very good point and we would like to thank the reviewer to bring this to our attention. We changed the wording as follows: "...", possibly indicating their lysosomal destination."

Reviewer #3 (Remarks to the Author):

My points have been sufficiently addressed and I support the publication of the manuscript.

R: We would like to thank this reviewer for the positive assessment of our work and support for publication of our manuscript.